# MeCo: Zero-Shot NAS with One Data and Single Forward Pass via Meinimum Eigenvalue of Correlation

**Tangyu Jiang**[†]
School of Artificial Intelligence
Beijing Normal University
jty@mail.bnu.edu.cn

**Haodi Wang**[†]
School of Artificial Intelligence
Beijing Normal University
whd@mail.bnu.edu.cn

**Rongfang Bie**[*]
School of Artificial Intelligence
Beijing Normal University
rfbie@bnu.edu.cn

## Abstract

Neural Architecture Search (NAS) is a promising paradigm in automatic architecture engineering. Zero-shot NAS can evaluate the network without training via some specific metrics called zero-cost proxies. Though effective, the existing zero-cost proxies either invoke at least one backpropagation or depend highly on the data and labels. To alleviate the above issues, in this paper, we first reveal how the Pearson correlation matrix of the feature maps impacts the convergence rate and the generalization capacity of an over-parameterized neural network. Enlightened by the theoretical analysis, we propose a novel zero-cost proxy called MeCo, which requires only one random data for a single forward pass. We further propose an optimization approach MeCo$_{opt}$ to improve the performance of our method. We design comprehensive experiments and extensively evaluate MeCo on multiple popular benchmarks. MeCo achieves the highest correlation with the ground truth (e.g., 0.89 on NATS-Bench-TSS with CIFAR-10) among all the state-of-the-art proxies, which is also fully independent of the data and labels. Moreover, we integrate MeCo with the existing generation method to comprise a complete NAS. The experimental results illustrate that MeCo-based NAS can select the architecture with the highest accuracy and a low search cost. For instance, the best network searched by MeCo-based NAS achieves 97.31% on CIFAR-10, which is 0.04% higher than the baselines under the same settings. Our code is available at https://github.com/HamsterMimi/MeCo.

## 1 Introduction

Deep Neural Networks (DNNs) have been ubiquitously adopted in various fields due to their ability in hierarchical feature extraction [1–5]. It is a common consensus that the architecture of DNN is essential in model performance. Unfortunately, the manual trial-and-error method is non-scalable and inefficient, making it infeasible to find the globally optimal structure. Thanks to the development of the Neural Architecture Search (NAS), it is possible to automatically select the network architecture with the best performance.

---

[*]Corresponding author.
[†]These authors contributed equally to this work.

37th Conference on Neural Information Processing Systems (NeurIPS 2023).

There are abundant works proposed to improve the performance of NAS. The most fundamental observation is to fully or partially train the candidate networks with specific data and select the network with the highest accuracy. Though effective, these multi-shot NAS methods are inevitably resource-consuming, thus, are non-trivial to be adopted in real-world applications. To address this issue, the one-shot NAS has been proposed in which a supernet is designed and only requires one-time full training [6]. In 2020, Mellor et al. [7] proposed the zero-shot NAS and realized an efficient approach in which a zero-cost proxy is proposed to evaluate the networks without training.

Despite the effectiveness of the existing zero-cost proxies, there are some critical issues that remain to be solved. (i) Almost all current proxies are established from the network gradients. As a consequence, computing these proxies requires at least one backpropagation, which is resource-consuming. (ii) The majority of the currently used proxies rely highly on the input data and labels. The reliance on the input samples underestimates the evaluation of the network's intrinsic characteristics, and the incorrect labels (which are commonly occurred in real-world datasets) may obfuscate the ranking results. (iii) Although various zero-cost proxies have been proposed, the performance of those proxies can still be promoted in network evaluation.

**Objectives.** The main objective of this paper is to design a novel zero-cost proxy with better performance than the current methods. The proposed proxy should be easily computed by a small number of data without backpropagation. Furthermore, to estimate the intrinsic trait of the architectures and avoid the label influence, our proxy should be label-independent.

**Our Contributions.** To this end, in this paper, we craft a novel zero-cost proxy for zero-shot NAS from a new perspective. The proposed proxy, called MeCo, outperforms the State-Of-The-Art (SOTA) zero-cost proxies in NAS and only requires one data sample for a single forward pass. Specifically, we harness a novel observation on the multi-channel convolutional layers and view them as the "multi-sample fully-connected layers with constraints". This insight permits MeCo to be computed by only one data. Moreover, we rigorously analyze how the Pearson correlation matrix of the feature maps impacts the training convergence rate and the generalization capacity of the networks, which theoretically proves the effectiveness of MeCo. Unlike the existing methods [7–9], MeCo is established upon each layer of the feature maps and hence requires only a single forward pass. Based on our design, we further propose an optimization approach $MeCo_{opt}$ to promote the performance of our method. The experimental results show that both MeCo and $MeCo_{opt}$ achieve a higher correlation with the network ground truth than the SOTA methods. For example, the Spearman correlation coefficients of MeCo on NATS-Bench-TSS with CIFAR-10 and CIFAR-100 are 0.89 and 0.88, respectively, which are 0.09 and 0.08 higher than the best baseline method. Our MeCo-based zero-shot NAS also acquires the highest accuracy among all the existing approaches, e.g., MeCo-based NAS selects the network with 97.31% accuracy, which is 0.04% higher than the SOTA work. In all, our major contributions are as follows.

- We theoretically analyze how the Pearson correlation matrix of the feature maps impact the training convergence rate and the generalization capacity of the networks, based on the characteristics of the multi-channel convolutional layers.

- We propose MeCo , a novel zero-cost proxy for zero-shot NAS that achieves better performance than SOTA methods. Through systematic analysis, we further proposed $MeCo_{opt}$ as an optimization approach. Our proxy is fully training-free and data-independent, which requires only one data sample with a single forward pass.

- We rigorously implement our MeCo and extensively design the experiments to evaluate its performance on several popular benchmarks (e.g., NAS-Bench-101, NATS-Bench-TSS, NATS-Bench-SSS, NAS-Bench-301, and Transbench-101) with diverse datasets and random data samples. The experimental results show that our zero-cost proxy outperforms the existing methods, and MeCo-based NAS can select the network with the highest accuracy.

## 2   Related work

### 2.1   Zero-shot NAS and zero-cost proxy

Zero-shot NAS uses specific metrics called zero-cost proxies to evaluate the networks. Unlike the traditional multi-shot and one-shot strategies [10, 6, 11–19], zero-shot NAS eliminates the training procedure thus significantly improves the efficiency. Various studies have been brought forward to

promote the quality of the zero-cost proxies, and most of them are based on the gradients. Mellor et al. [7] firstly evaluated the performance of the initialized networks using the activation overlaps between data points. Abdelfattah et al. [8] proposed a series of parameter-pruning based proxies including *snip* [20], *grasp* [21], *synflow* [22], and *fisher* [23]. Recently, Li et al. [9] proposed *ZiCo*, which firstly works better than the parameter amounts. These approaches need at least one backpropagation and rely on the data labels to compute the proxies. Another line of work [24–26] is based on the theory of deep learning such as NTK, which is independent of the labels yet still needs backpropagation. For example, Chen et al. [24] utilized the condition number of NTK to evaluate the trainability of the networks. Finally, there are a few works that do not require backpropagation or data labels. For instance, Lin et al. [27] proposed Zen-Score, in which they design an efficient zero-cost proxy with Gaussian random inputs. However, Zen-Score requires multiple forward passes while our MeCo only needs a single forward pass.

## 2.2 Over-parameterized networks

Over-parameterized networks have received a lot of attention due to their outstanding effect and ease of optimization. Jacot et al. [28] demonstrated that the training dynamic of an infinite-width network follows a kernel called NTK [29, 30]. For finite-width networks, the training dynamic can be illustrated by a gram matrix [31, 32]. Further, Du et al. [31] proved that the loss of an over-parameterized network could converge to a global minimum [33–35], and the training convergence rate could be characterized by the minimum eigenvalue of the gram matrix. Some works also discuss the optimization of the training convergence rate [36] and the topological properties of DNNs [37].

## 3 MeCo: minimum eigenvalue of correlation on feature maps

### 3.1 Preliminaries

**Notations.** We define $[n] = \{1, 2, \ldots, n\}$. The lower and uppercase bold font represent vectors and matrices, respectively, e.g., $\mathbf{x}$ is a vector with entry $x_i$, and $\mathbf{M}$ is a matrix with entry $[\mathbf{M}]_{ij}$. The minimum eigenvalue of $\mathbf{M}$ is denoted as $\lambda_{\min}(\mathbf{M})$. We define $\mathbf{A} \circ \mathbf{B}$ as Hadamard product between two same-sized matrices $\mathbf{A}$ and $\mathbf{B}$. $\mathbf{1}_{m \times n}$ represents a $m \times n$ matrix filled by ones and $\| \cdot \|_2$ is used to represent the $l_2$ norm of a vector. $N(\mathbf{0}, \mathbf{I})$ and $U\{S\}$ represent the standard Gaussian distribution and uniform distribution over a set $S$, respectively. We denote by $\mathbf{X} = \{(\mathbf{x}_i, y_i) | \mathbf{x}_i \in \mathbb{R}^{d \times 1}, y_i \in \mathbb{R}, i \in [n]\}$ the training set, where $\mathbf{x}_i$ and $y_i$ represent the $i$-th data and label. $\mathbb{I}\{\cdot\}$ is defined as the indicator function that demonstrates the event occurrence, such that for event $\mathcal{A}$, $\mathbb{I}\{\mathcal{A}\} = 1$ if and only if $\mathcal{A}$ happened, otherwise it equals to 0.

For input $\mathbf{x} \in \mathbb{R}^{d \times 1}$, weight vector $\mathbf{w} \in \mathbb{R}^{d \times 1}$ in the weight matrix $\mathbf{W} \in \mathbb{R}^{d \times m}$, and output weight $\mathbf{a} \in \mathbb{R}^{m \times 1}$, we denote $f(\mathbf{W}, \mathbf{a}, \mathbf{x})$ as a neural network with a single hidden layer such that

$$f(\mathbf{W}, \mathbf{a}, \mathbf{x}) = \frac{1}{\sqrt{m}} \sum_{r=1}^{m} a_r \sigma(\mathbf{w}_r^T \mathbf{x}) \tag{1}$$

where $\sigma$ is the activation function. In this paper, we mainly consider the ReLU function due to its effectiveness, i.e., $\sigma(z) = z\mathbb{I}\{z > 0\}$. Given a training set $\mathbf{X}$, the optimization goal is to minimize the empirical risk loss function

$$L(\mathbf{W}, \mathbf{a}) = \sum_{i=1}^{n} \frac{1}{2}(f(\mathbf{W}, \mathbf{a}, \mathbf{x}_i) - y_i)^2 \tag{2}$$

We follow the definitions in [31] and define the matrices $\mathbf{H}(t)$ and $\mathbf{H}^\infty$ as follows.

**Definition 1** (Gram Matrix). *For a neural network with a single hidden layer, the gram matrix* $\mathbf{H}(t) \in \mathbb{R}^{n \times n}$ *induced by the ReLU activation function on a training set* $\mathbf{X} := \{(\mathbf{x}_i, y_i)\}_{i=1}^{n}$ *with entry* $[\mathbf{H}(t)]_{ij}$ *is defined as:*

$$[\mathbf{H}(t)]_{ij} = \frac{1}{m} \sum_{r=1}^{m} \mathbf{x}_i^T \mathbf{x}_j \mathbb{I}\{\mathbf{w}_r^T(t)\mathbf{x}_i \geq 0, \mathbf{w}_r^T(t)\mathbf{x}_j \geq 0\} \tag{3}$$

*where* $\mathbf{w}_r(t)$ *is a vector that depends on* $t$. *We further construct* $\mathbf{H}^\infty$ *with entry* $[\mathbf{H}^\infty]_{ij}$ *such that*

$$[\mathbf{H}^\infty]_{ij} = \mathbb{E}_{\mathbf{w} \sim N(\mathbf{0}, \mathbf{I})}[\mathbf{x}_i^T \mathbf{x}_j \mathbb{I}\{\mathbf{w}^T \mathbf{x}_i \geq 0, \mathbf{w}^T \mathbf{x}_j \geq 0\}] \tag{4}$$

*We denote* $\lambda_0 := \lambda_{\min}(\mathbf{H}^\infty)$.

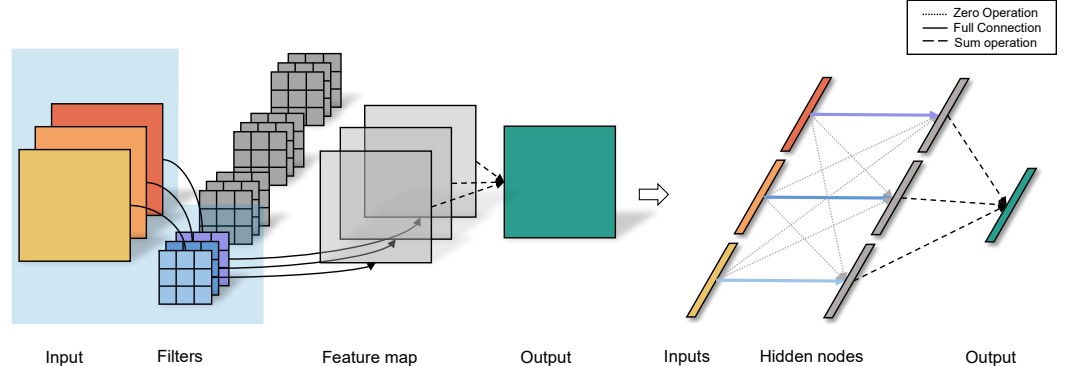

Input    Filters        Feature map    Output        Inputs    Hidden nodes    Output

Figure 1: Conversion between the multi-channel convolution and multi-sample fully-connected operation. The input size is $3 \times w \times h$ and each filter size is $3 \times 3 \times 3$. Each input channel can be flattened with size $d \times 1$, $d = w \times h$. We collect the three flattened samples to obtain the output with constrained fully-connected operations (dot lines for zero operations and solid lines for full connection). The final output is computed by a sum operation (dashed lines) from the hidden nodes.

**Remark 1.** *Gram matrix $\mathbf{H}(t)$ reflects the prediction dynamic at the $t$-th iteration of the network training. Du et al. [31] proved that for all $t \geq 0$, $\|\mathbf{H}(t) - \mathbf{H}^\infty\|_2 \to 0$ as $m \to \infty$. Moreover, the gram matrix $\mathbf{H}^\infty$ has the following key property: if $\mathbf{x}_i \nparallel \mathbf{x}_j, \forall i \neq j$, then $\lambda_0 > 0$. The proof of this property can be found in Theorem 3.1 in [31].*

**Definition 2** (Pearson Correlation Matrix). *$P(\mathbf{X})$ is a Pearson correlation matrix with the $(i,j)$-th entry $[P(\mathbf{X})]_{ij}$ as*

$$[P(\mathbf{X})]_{ij} = \rho(\mathbf{x}_i, \mathbf{x}_j) = \frac{\mathbb{E}[(\mathbf{x}_i - \mu_{\mathbf{x}_i})(\mathbf{x}_j - \mu_{\mathbf{x}_j})]}{\sigma_{\mathbf{x}_i} \sigma_{\mathbf{x}_j}} \tag{5}$$

*where $\mu_{\mathbf{x}}$ and $\sigma_{\mathbf{x}}$ are the mean and standard deviation of $\mathbf{x}$, respectively.*

**Remark 2.** *Pearson correlation matrix $P(\mathbf{X})$ is closely related to $\mathbf{H}(t)$. For instance, suppose $\mathbf{x}_1, \mathbf{x}_2 \sim N(\mathbf{0}, \mathbf{I})$, $\mathbf{x}_i \in \mathbb{R}^{d \times 1}$ and $\mathbf{w}_r^T(t)\mathbf{x}_i \geq 0$, for $r \in [m]$ and $i \in [2]$, then $[\mathbf{H}(t)]_{12} = (d-1)[P(\mathbf{X})]_{12}$. We will use this characteristic to approximate the training convergence rate and generation capacity of the networks when designing our proxy.*

## 3.2 The construction of MeCo

To present the theoretical analysis, we first approximate the Convolutional Neural Network (CNN) to an over-parameterized Neural Network (NN) and then craft our proxy.

### 3.2.1 Back to convolution

The convolutional layers are the fundamental parts of CNN. Suppose we have the input $\mathbf{x}_{\text{in}} \in \mathbb{R}^{c_{\text{in}} \times w \times h}$ and the filter $\mathbf{w} \in \mathbb{R}^{c_{\text{in}} \times k \times k}$. We set the stride size to one and retain the size of the feature maps via zero padding, then the convolutional layer can be formally expressed as:

$$[\mathbf{x}_{\text{out}}]_{i,j} = \sum_{c=1}^{c_{\text{in}}} \sum_{a=-p}^{p} \sum_{b=-p}^{p} [\mathbf{w}^c]_{a+p+1, b+p+1} \cdot [\mathbf{x}_{\text{in}}^c]_{a+i, b+j} \tag{6}$$

where $\mathbf{x}_{\text{in}}^c$ (*resp.* $\mathbf{w}^c$) represents the $c$-th channel of $\mathbf{x}_{\text{in}}$ (*resp.* $\mathbf{w}$), $p = (k-1)/2$, $i \in [w]$, $j \in [h]$. In this paper, we observe that $\mathbf{x}_{\text{out}}$ and $\mathbf{x}_{\text{in}}^c$ can be flattened to one-dimensional vectors $\tilde{\mathbf{x}}_{\text{out}} \in \mathbb{R}^{d \times 1}$ and $[\tilde{\mathbf{x}}_{\text{in}}^c] \in \mathbb{R}^{d \times 1}$, where $c \in [c_{\text{in}}]$, $d = w \times h$. More concretely, we have the following transformations:

$$\tilde{\mathbf{x}}_{\text{out}} = \sum_{c=1}^{c_{\text{in}}} \mathbf{A}_c \sigma((\mathbf{B}_c \circ \mathbf{W})^T \tilde{\mathbf{x}}_{\text{in}}^c) \tag{7}$$

$$s.t. \quad \mathbf{A}_c \in \mathbb{R}^{d \times d_h}, [\mathbf{A}_c]_{ij} = \mathbb{I}\{j = (c-1)d + i\}$$

$$\mathbf{B}_c \in \mathbb{R}^{d_h \times d}, [\mathbf{B}_c]_{ij} = \mathbb{I}\{(c-1)d < i \leq cd\}, \tag{8}$$

$$\mathbf{B}_c \circ \mathbf{W} \text{ satisfies weight sharing constraints}$$

where $d_h = c_\text{in} \times d$, $\mathbf{W} \in \mathbb{R}^{d \times d_h}$ is the weight matrix. Hence, the convolutional layer in CNN is equivalent to a fully-connected layer with constraints, i.e., shown in Equation 8. To further simplify the problem, we relax the second constraint to $\mathbf{B}_c = \mathbf{1}_{d_h \times d}$ and ignore the last constraint. As a consequence, we obtain a multi-sample fully-connected network in which each flattened channel $\tilde{\mathbf{x}}_\text{in}^c$ is regarded as an independent data sample, and the total number of samples is $c_\text{in}$. To better illustrate our insight, we present an example of the conversion procedure in Figure 1.

### 3.2.2 Over-parameterized neural networks

The over-parameterized NNs are competitive in hierarchical feature extraction due to a large number of parameters. One of the typical architectures is the networks with wide hidden layers, which is proved to be tractable in training [38]. In the previous subsection, we convert a multi-channel convolutional layer to a multi-sample fully-connected layer with constraints. We further argue that if the number of hidden nodes in the transformed fully-connected layer is large enough, then it can be viewed as an over-parameterized NN layer. Therefore, the characteristics of the over-parameterized NN can be transferred to CNN. To this end, we present the following theorem [31] about the convergence rate of the fully-connected NN with a single hidden layer as follows.

**Theorem 1.** *If gram matrix $\mathbf{H}^\infty \succ 0$, $\|\mathbf{x}_i\|_2 = 1$, $|y_i| < C$ for some constant $C$ and $i \in [n]$, hidden nodes $m = \Omega\left(\frac{n^6}{\lambda_0^4 \delta^3}\right)$, and i.i.d. initialize $\mathbf{w}_r \sim N(\mathbf{0}, \mathbf{I})$, $a_r \sim U\{[-1, 1]\}$ for $r \in [m]$, then with probability at least $1 - \delta$ over the initialization, the following inequality holds:*

$$\|f(\mathbf{W}(t), \mathbf{a}, \mathbf{X}) - \mathbf{y}\|_2^2 \leq \exp(-\lambda_0 t)\|f(\mathbf{W}(0), \mathbf{a}, \mathbf{X}) - \mathbf{y}\|_2^2 \tag{9}$$

**Remark 3.** *This inequality shows that $\lambda_0$ positively affects the training convergence rate of the network. We further point out that the convergence rate of the network is independent of the label $\mathbf{y}$, which makes it possible to measure the network performance with only a single forward pass. Based on this observation, we establish our proxy on the intermediate feature maps to measure the performance of the network.*

Based on Theorem 1 and Remark 2, we further analyze the relationship between the minimum eigenvalue of the matrix $P(\mathbf{X})$ and the training convergence rate of the over-parameterized network. We summarize our results in the following theorem:

**Theorem 2.** *Suppose $f$ is an NN with a single hidden layer and ReLU activation function. Assume $\mathbf{X} \in \mathbb{R}^{d \times n}$, $\mathbf{w}(0) \sim N(\mathbf{0}, \mathbf{I})$, $P(\mathbf{X}) \succ 0$, $p_0 := \lambda_{\min}(P(\mathbf{X}))$, and hidden nodes $m = \Omega\left(\frac{n^6 d^2}{\lambda_0^4 \delta^3}\right)$, then the following formula holds with probability at least $1 - \delta$ over the initialization*

$$\|f(\mathbf{W}(t), \mathbf{a}, \mathbf{X}) - \mathbf{y}\|_2^2 \leq \exp(-cp_0 t)\|f(\mathbf{W}(0), \mathbf{a}, \mathbf{X}) - \mathbf{y}\|_2^2 \tag{10}$$

*where $c$ is a constant depending on $m$, and $d$.*

*Proof sketch.* The key of the proof is to find the relationship between $\lambda_0$ and $p_0$ by Hoeffding's inequality and Weyl inequalities [39]. We provide the full proof in Appendix A.1. Theorem 2 shows that $p_0$ can reflect the convergence rate of the network to a certain extent. The advantage of $P(\mathbf{X})$ over $\mathbf{H}(t)$ is that its computation only depends on the data. Therefore, it is feasible to estimate the training convergence rate of the network only through the feature maps.

Other than the convergence rate, we also analyze the relationship between $p_0$ and the *generalization capacity* of the over-parameterized NN. We present the results in the following theorem:

**Theorem 3.** *For an over-parameterized neural network with the loss on the testing set as $L(\mathbf{W})$. Let $\mathbf{y} = (y_1, ..., y_N)^T$, and $\gamma$ be the step of SGD, $\gamma = \kappa C_1 \sqrt{\mathbf{y}^T (\mathbf{H}^\infty)^{-1} \mathbf{y}}/(m\sqrt{N})$ for some small enough absolute constant $\kappa$. Under the assumption of Theorem 2, for any $\delta \in (0, e^{-1}]$, there exists $m^*(\delta, N, \lambda_0)$, such that if $m \geq m^*$, then with probability at least $1 - \delta$, we have*

$$\mathbb{E}[L(\mathbf{W})] \leq \mathcal{O}(C'\sqrt{\frac{\mathbf{y}^T \mathbf{y}}{p_0 N}}) + \mathcal{O}(\sqrt{\frac{\log(1/\delta)}{N}}) \tag{11}$$

*where $C, C', \delta$ are constants.*

*Proof sketch.* The proof of the above theorem derives from the Corollary 3.10 of [40] and Section D.2 of [26]. We present the detailed proof in Appendix A.2.

### 3.2.3 New zero-cost proxy: MeCo

The above theorem and analysis indicate the following key insights:

- A multi-channel CNN layer can be viewed as a multi-sample over-parameterized NN layer.
- The minimum eigenvalue of $P(\mathbf{X})$ positively affects the training convergence rate and the generalization capacity of an NN, which is independent of the labels.

Enlightened by these observations, we harness the Minimum eigenvalue of the Pearson Correlation matrix upon each layer of the feature maps to craft our novel zero-cost proxy called MeCo, such that

**Definition 3** (MeCo). *Assume the NN $f(\cdot; \theta)$ has a total of $D$ layers, then* MeCo *is defined as*

$$\text{MeCo} := \sum\nolimits_{i=1}^{D} \lambda_{\min}(P(f^i(\mathbf{X}; \theta))) \tag{12}$$

*where $f^i(\mathbf{X}; \theta)$ represents the $i$-th feature map with the initialized parameters $\theta$ on dataset $\mathbf{X}$.*

**Remark 4.** *Note that the Pearson correlation matrix contains the process of data normalization, thus our* MeCo *eliminates the deviation caused by the data, which is more conducive to discovering the essence of the network architectures.*

## 4  Experiments and evaluations

We evaluate MeCo from three aspects: (i) The correlation with the ground truth and comparison with the baseline proxies; (ii) The dependency on data/labels; (iii) Performance of MeCo-based NAS scheme. We first describe the experimental configurations and then give the detailed performance.

### 4.1  Experimental configurations

**Hardware.** We fully implement our MeCo in Python. We evaluate MeCo on a desktop with an Inter Core i7-12700F CPU and GeForce RTX 3090.

**Benchmarks.** We use several popular benchmarks in NAS. (i) *NATS-Bench-TSS* [41, 42] contains 15,625 CNN architectures with different topologies. We demonstrate the performance of the zero-cost proxied on NATS-Bench-TSS with CIFAR-10 [43], CIFAR-100 [43], ImageNet16-120[44], and three extra datasets [45] in Appendix E.1. (ii) *NATS-Bench-SSS* [42] contains 32,768 CNN architectures with the same topology but with different numbers of channels. We show the results on NATS-Bench-SSS with CIFAR-10, CIFAR-100, and ImageNet16-120. (iii) *NAS-Bench-301* [46] is a surrogate NAS benchmark for the DARTS search space. We illustrate the results with CIFAR-10 and the extra three datasets in Appendix E.1. (iv) *TransBench-101* [47] is a benchmark composed of two subsets, i.e., TransBench-101-Micro and TransBench-101-Macro. The former is evaluated with 10 datasets and the latter with 7 datasets. We present these results in Appendix E.3. (v) *NAS-Bench-101* [48] includes 423,624 cell-based CNN architectures, which are trained and evaluated on the CIFAR-10 dataset and shown in Appendix E.2. (vi) *AutoFormer* [49] and *MobileNet OFA* [50]. The description and results are shown in Appendix E.4.

**Baselines.** The baselines for zero-cost proxies include the number of parameters #Param, grasp [21], fisher [23], snip [20], synflow [22], jacov [7], grad_norm [8], NTK [24], zen [27], NASWOT [7], KNAS [51], NASI [52], GradSign [53], and ZiCo [9]. Specific forms of these proxies are provided in Appendix B. We adopt the Spearman rank correlation coefficient $\rho$ between zero-cost proxies and the dataset ground truth to measure their performance.

### 4.2  Performance of our MeCo and comparisons

We evaluate $\rho$ between MeCo and the ground truth, and compare it with the baseline zero-cost proxies on various benchmarks. We present the results on NATS-Bench-TSS and NATS-Bench-SSS in Table 1, and the results on NAS-Bench-301 are provided in Table 2. We run MeCo for ten times with different inputs and present the mean and standard deviation.

**NATS-Bench-TSS.** We run our MeCo and all baseline proxies on 15,625 networks of NATS-Bench-TSS with three public datasets. As we can see in Table 1, MeCo achieves 0.894±0.003, 0.883±0.005,

Table 1: Spearman correlation coefficients $\rho$ of proxies on NATS-Bench-TSS and NATS-Bench-SSS

| Approach | NATS-Bench-TSS | | | NATS-Bench-SSS | | |
|---|---|---|---|---|---|---|
| | CIFAT-10 | CIFAR-100 | ImageNet16 | CIFAT-10 | CIFAR-100 | ImageNet16 |
| grasp | 0.39 | 0.46 | 0.45 | -0.13 | 0.01 | 0.42 |
| fisher | 0.40 | 0.46 | 0.42 | 0.44 | 0.55 | 0.47 |
| grad_norm | 0.42 | 0.49 | 0.47 | 0.51 | 0.49 | 0.67 |
| snip | 0.43 | 0.49 | 0.48 | 0.59 | 0.62 | 0.76 |
| synflow | 0.74 | 0.76 | 0.75 | **0.81** | 0.80 | 0.57 |
| #Param | 0.72 | 0.73 | 0.69 | 0.72 | 0.73 | 0.84 |
| NWOT | 0.77 | 0.80 | 0.77 | 0.45 | 0.43 | 0.42 |
| jacov | 0.73 | 0.70 | 0.70 | 0.30 | 0.13 | 0.30 |
| NTK | 0.76 | 0.75 | 0.72 | 0.34 | 0.29 | 0.28 |
| zen | 0.38 | 0.36 | 0.40 | 0.69 | 0.71 | 0.87 |
| KNAS | 0.20 | 0.35 | 0.42 | 0.25 | 0.12 | 0.32 |
| NASI | 0.44 | 0.43 | 0.63 | 0.17 | 0.04 | 0.20 |
| GradSign | 0.77 | 0.79 | 0.78 | 0.21 | 0.16 | 0.04 |
| ZiCo | 0.80 | 0.81 | 0.79 | 0.73 | 0.75 | **0.88** |
| MeCo (**Ours**) | **0.894**±**0.003** | **0.883**±**0.005** | **0.845**±**0.004** | -0.79±0.01 | **-0.87**±**0.01** | -0.86±0.02 |
| MeCo$_{opt}$ (**Ours**) | **0.901**±**0.002** | **0.890**±**0.003** | **0.850**±**0.003** | **0.89**±**0.002** | **0.83**±**0.004** | **0.89**±**0.003** |

and 0.845±0.004 correlation on CIFAR-10, CIFAR-100, and ImageNet16-120, respectively. The variation shows that our method is stable and independent of the input samples. The Spearman correlation coefficients are significantly higher than the baselines, e.g., the SOTA proxy ZiCo [9] only achieves 0.80, 0.81, 0.79 on three datasets, which is 0.06 to 0.09 lower than ours. Note that ZiCo is the first proxy that is proved to be consistently better than the naive proxy #Param and requires a batch of input data. On the other hand, we surpass all the previous works and only use one data sample. The high correlation coefficients demonstrate the effectiveness of MeCo, which is consistent with the theoretical analysis. To make our results more intuitive, we sample 2,000 networks randomly and present the relationship between MeCo and the ground truth on three public datasets. The results are shown in Figure 4(a) in the Appendix E, where the trend in the scatter charts effectively proves that our zero-cost proxy has a highly strong correlation with the architecture quality.

**NATS-Bench-SSS.** The results of MeCo on 32,768 networks of NATS-Bench-SSS are summarized in Table 1. Among all the datasets, MeCo shows the highest correlation with the ground truth on CIFAR-100, i.e., $\rho = -0.87 \pm 0.01$, which is 0.07 higher than the best baseline synflow. MeCo also achieves stable and competitive results on CIFAR-10 and ImageNet16-120 with one data as input. We find that MeCo is highly *negatively* correlated with the test accuracy, which is somehow counterintuitive. In fact, MeCo is sensitive to the number of channels (#channels) that cause this phenomenon. Specifically, the channels are viewed as the data samples as described in Section 3.2.1, This leads to the fact that the more channels contained, the larger the Pearson matrix, which will lead to **smaller** $p_0$. On the other hand, larger #channels will normally lead to higher accuracy of the architectures (yet lower

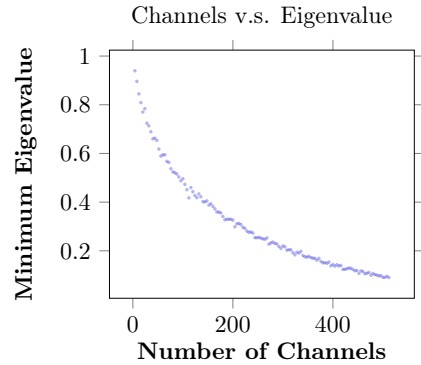

Figure 2: The influence of different channel numbers of data on the minimum eigenvalue.

convergence rate). Thus, the variation of #channels in NATS-Bench-SSS(chosen from 8 to 64) leads to a negative correlation on this benchmark. In NATS-Bench-TSS, however, all the networks share the same #channels at each stage, thus MeCo does not show a negative correlation. To verify this, we generate random data with different numbers of channels and calculated $p_0$ respectively, and demonstrate the results in Figure 2. We also sample 2,000 networks randomly and present the correlation of MeCo v.s. test accuracy in Figure 4(b) in the Appendix E.

**NATS-Bench-301**. We demonstrate the results of MeCo and the baseline proxies on NATS-Bench-301 with CIFAR-10 in Table 2. It can be shown from the results that MeCo and MeCo_{opt} also achieve the highest correlation with the testing accuracy on NATS-Bench-301, which is 0.04-0.05 higher than the SOTA approach ZiCo. More experimental results of MeCo can be found in Appendix E

Table 2: $\rho$ between zero-cost proxies and test accuracy on NAS-Bench-301 with CIFAR-10

| Dataset | MeCo | MeCo_{opt} | Baselines | | | | | | | | | | |
|---------|------|------------|-------|--------|-----------|------|---------|--------|---------|------|------|------|------|
| | | | grasp | fisher | grad_norm | snip | synflow | l2_norm | #Param | zen | jacov | nwot | ZiCo |
| CIFAR-10 | **0.7±0.01** | **0.71±0.01** | 0.34 | -0.28 | -0.04 | -0.05 | 0.18 | 0.45 | 0.46 | 0.43 | -0.04 | 0.47 | 0.66 |

## 4.3 Dependency on data

MeCo is designed to be fully data-independent due to the characteristics of Pearson correlation matrix (Theorem 2). To further illustrate this attribute, we generate a random dataset and use it as the input of MeCo and NTK [24], which is also independent of the data and labels. We compute the Spearman correlation coefficients between MeCo and the test accuracy on three public datasets, and compare them with the baseline approach. All the results for MeCo is ran ten times with random inputs.

More concretely, the random data is generated following a Gaussian distribution and fed into the initialized network. Note that the baseline method requires a batch of random data as input, while MeCo only uses one. We present the results for NATS-Bench-TSS and NATS-Bench-SSS in Table 3.

The results show that even if we use random data as the inputs, MeCo is still able to maintain a high correlation with the corresponding ground truth. For example, $\rho = 0.87$ on NATS-Bench-TSS with CIFAR-10 and $\rho = -0.93$ on NATS-Bench-SSS with ImageNet16-120, which are significantly higher than the data-independent proxy NTK. Note that most existing proxies rely on the gradients and backpropagation. Hence they cannot utilize the random input data for architecture evaluation, such as ZiCo and jacov.

Table 3: Spearman correlation coefficients of MeCo and NTK with random data on NATS-Bench-TSS and NATS-Bench-SSS

| Dataset | NTAS-Bench-TSS | | NATS-Bench-SSS | |
|---------|------|---------|------|---------|
| | MeCo | NTK[24] | MeCo | NTK[24] |
| **CIFAR-10** | 0.87±0.001 | 0.78 | -0.87±0.000 | 0.23 |
| **CIFAR-100** | 0.85±0.001 | 0.79 | -0.88±0.001 | 0.16 |
| **ImageNet16-120** | 0.83±0.001 | 0.76 | -0.94±0.000 | 0.20 |

## 4.4 NAS with MeCo

We integrate our MeCo with the existing search space and generation strategy to comprise a complete NAS method. Specifically, we use NATS-Bench-TSS, DARTS-CNN [6], and MobileNet V3 [54] as the search spaces to demonstrate the performance. We use the Zero-Cost-PT [55] as the generation strategy. The algorithm of MeCo-based NAS and the results on MobileNet V3 are presented in Appendix C and F.3, respectively.

**NATS-Bench-TSS.** We directly evaluate all networks in NATS-Bench-TSS using MeCo and baseline proxies. The average test accuracy of the Top-10 networks on three datasets is summarized in Table 12 in Appendix F.1. As shown in the table, the average test accuracy on CIFAR-10 of MeCo is $0.08\%$ higher than the best baseline

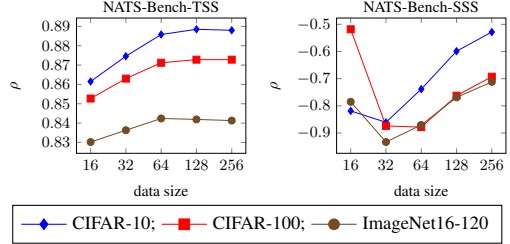

Figure 3: Spearman correlation coefficients of MeCo with different data size on NATS-Bench-TSS and NATS-Bench-SSS, respectively.

ZiCo. Moreover, we further search the networks using Zero-Cost-PT with different proxies and run the search algorithm three times. The networks we searched have the highest average test accuracy on three public datasets, that is, 93.76%, 71.11%, and 41.44% on CIFAR-10, CIFAR-100, and ImageNet16-120, respectively. The results are provided in Table 13 in Appendix F.1.

**DARTS-CNN.** We compare our MeCo-based NAS with multi-shot, one-shot, and zero-shot NAS schemes in DARTS-CNN spaces. All the NAS schemes are executed on CIFAR-10 and CIFAR-100

datasets, respectively, to compare the Top-1 errors and the search cost. For zero-shot NAS, we use Zero-Cost-PT as the generation strategy due to its effectiveness. The results on CIFAR-10 are summarized in Table 4, where we denote multi-shot, one-shot, and zero-shot as MS, OS, and ZS, respectively. We present the CIFAR-100 results in Appendix F.2. The results demonstrate that our MeCo-based NAS outperforms the existing baseline methods. More concretely, for zero-shot methods, MeCo outperforms all the baselines for 0.04% to 0.43% on Top-1 error with a similar level of search cost under the same generation strategy. For training-based methods, MeCo-based NAS achieves competitive accuracy but with much less search cost. For instance, our approach requires only 0.08 GPU days and obtains 97.31% accuracy, while DARTS-PT costs $10\times$ GPU days than ours with 0.08% higher accuracy.

Table 4: Comparisons of MeCo-based NAS with baselines using DARTS-CNN and CIFAR-10

| Approach | Test Error (%) | Search Cost (GPU Days) | Params (M) | Method |
|---|---|---|---|---|
| AmoebaNet-A [56] | $3.34 \pm 0.06$ | 3150 | 3.2 | MS |
| PNAS [57] | $3.41 \pm 0.09$ | 225 | 3.2 | MS |
| DARTS (1st) [6] | $3.00 \pm 0.14$ | 1.5 | 3.3 | OS |
| DARTS-PT[58] | $2.61 \pm 0.08$ | 0.8 | 3.0 | OS |
| SDARTS-RS [59] | $2.67 \pm 0.03$ | 0.4 | 3.4 | OS |
| SGAS (Cri.1)[60] | $2.66 \pm 0.24$ | 0.8 | 3.7 | OS |
| Eigen-NAS[26] | 7.4 | - | - | ZS |
| TE-NAS [24] | $2.63 \pm 0.064$ | 0.05 | 3.8 | ZS |
| Zero-Cost-PT$_{synflow}$[22] | $2.96 \pm 0.11$ | 0.03 | 5.1 | ZS |
| Zero-Cost-PT$_{fisher}$[23] | $3.12 \pm 0.16$ | 0.05 | 2.5 | ZS |
| Zero-Cost-PT$_{grasp}$ [21] | $2.73 \pm 0.10$ | 0.1 | 3.3 | ZS |
| Zero-Cost-PT$_{jacov}$[7] | $2.88 \pm 0.15$ | 0.04 | 3.5 | ZS |
| Zero-Cost-PT$_{snip}$ [20] | $2.90 \pm 0.03$ | 0.04 | 4.0 | ZS |
| Zero-Cost-PT$_{NTK}$ [24] | $2.89 \pm 0.09$ | 0.21 | 4.1 | ZS |
| Zero-Cost-PT$_{ZiCo}$[9] | $2.80 \pm 0.03$ | 0.04 | 5.1 | ZS |
| **Zero-Cost-PT$_{MeCo}$(Ours)** | $\mathbf{2.69 \pm 0.05}$ | **0.08** | 4.2 | ZS |

We remark that the experimental results in this section demonstrate the following properties of MeCo: (i) **High correlation with the ground truth;** (ii) **Requies only one data without labels for a single forward pass.** That is because MeCo is established upon the observations over the multi-channel CNN layers. Moreover, the minimum eigenvalue of the Pearson correlation matrix is independent of the data labels, and thus we exert MeCo on the feature maps instead of the gradients.

## 5 Discussion and Optimization

### 5.1 Optimization method MeCo$_{opt}$

In the previous sections, we illustrated the theoretical basis of MeCo and evaluated our proposed proxy on various benchmarks. However, as described in Section 4.2, the negative correlation on NATS-Bench-SSS and some other tasks (e.g., Transbench-101 in the appendix) reflect that MeCo is highly sensitive to the number of channels. Although this phenomenon does not contradict the theoretical results or undermine the effectiveness of MeCo, it might be problematic when $p_0 = 0$. To this end, we present an optimization method on top of MeCo to alleviate the channel-sensitive trait.

Specifically, for convolutional layers, $p_0$ is strictly greater than zero if for $\forall i \neq j$, $x_i \nparallel x_j$, and $c < w \times h$, where $c$ is the number of channels, $w, h$ are the size of the inputs. Thus, in real-world applications, MeCo might lose efficacy during the down-sampling procedure. To address this issue, instead of flattening all the channels of the feature map as described in Section 3.2.1, we randomly sample a *fixed* number of channels and flatten them as matrix $P'$. We then compute the final result by multiplying the minimum eigenvalue of $\mathbf{P}'$ with a channel weight, such that

$$\text{MeCo}_{opt} := \sum_{i=1}^{D} \frac{c^{(i)}}{n} \cdot \lambda_{\min}(P') \tag{13}$$

where $c^{(i)}$ is the number of channels in the $i$-th layer, and $n$ is the fixed sampling numbers. The high-level idea of this optimization is to limit the dimension of the Pearson correlation matrix by constraining #channels. Instead of computing upon a large matrix, we calculate the minimum eigenvalue upon a fixed-sized matrix and then enlarge it with corresponding constants. Note that the bonus of this optimization is that the time cost can be controlled because the matrix dimension is significantly reduced compared to the original one.

To show the effectiveness of MeCo$_{opt}$, we calculate the Spearman correlation coefficients of MeCo$_{opt}$ on NATS-Bench-TSS, NATS-Bench-SSS (Table 1), NATS-Bench-301 (Table 2 and Table 8), and Transbench-101 (Table 10 and Table 11). All the experiments are conducted under $n = 8$. The results demonstrate that MeCo$_{opt}$ effectively solves the negative correlation issue on multiple benchmarks, e.g., NATS-Bench-SSS and TransBench-101. Meanwhile, the majority of the positive correlations are promoted up to 0.23, and the remaining results are not impacted.

## 5.2 Ablation study

**Sample size**. We consider the effect of the sample size on the performance of MeCo. We select the size of data from $\{16, 32, 64, 128, 256\}$, respectively, and calculate the Spearman rank correlation coefficient $\rho$ between MeCo and the test accuracy of the networks on three public datasets. The results are summarized in Figure 3. For NATS-Bench-TSS, $\rho$ increases rapidly before size 64 and grows slowly after that. The maximum value is acquired at size 128, which is slightly higher than size 64. On the other hand, the overall fluctuation is relatively large on NATS-Bench-SSS. The Spearman correlation coefficients are low when the size is smaller than 32, and reach higher at 32. Then $\rho$ reduces as the sample size grows larger. The ablation results show that the most appropriate size of random data is around 32 and 64, which is also adopted in our settings.

**Sample numbers**. We explore the performance of MeCo regarding the sample numbers that are used as the inputs. We randomly choose a different number of samples to evaluate the architectures in the entire NATS-Bench-TSS and NATS-Bench-SSS, respectively. The results are shown in Figure 5. It can be demonstrated from the figure that MeCo is relatively stable when varying the number of samples. The performance of MeCo fluctuates around 0.01 throughout the experiments. These results support that MeCo only requires one data sample.

**Fixed channel numbers** $n$. We conduct an ablation study on $n$ v.s. MeCo$_{opt}$ with NATS-Bench-TSS and NATS-Bench-SSS to illustrate the best choice of $n$. The results are shown in Table 5.

Table 5: Spearman correlation coefficients $\rho$ of proxies on NATS-Bench-TSS and NATS-Bench-SSS

| Approach | NATS-Bench-TSS | | | NATS-Bench-SSS | | |
|---|---|---|---|---|---|---|
| | CIFAT-10 | CIFAR-100 | ImageNet16 | CIFAT-10 | CIFAR-100 | ImageNet16 |
| $n = 4$ | $0.87 \pm 0.003$ | $0.88 \pm 0.002$ | $0.84 \pm 0.005$ | $0.88 \pm 0.005$ | $0.82 \pm 0.003$ | $0.84 \pm 0.003$ |
| $n = 6$ | $0.88 \pm 0.004$ | $0.88 \pm 0.001$ | $0.84 \pm 0.004$ | $0.88 \pm 0.007$ | $0.83 \pm 0.002$ | $0.85 \pm 0.006$ |
| $n = 8$ | $0.90 \pm 0.002$ | $0.89 \pm 0.003$ | $0.85 \pm 0.003$ | $0.89 \pm 0.002$ | $0.83 \pm 0.004$ | $0.89 \pm 0.003$ |

The results show that with $n$ growing larger, MeCo$_{opt}$ increases by 0.1 to 0.5. We set $n = 8$ in our experiments to obtain the best performance.

## 6 Conclusion

In this work, we propose a novel feature-based zero-cost proxy called MeCo and its optimization method MeCo$_{opt}$. Unlike the existing methods, our zero-cost proxies require only one data for a single forward pass. Specifically, we theoretically approximate a multi-channel convolution layer to an over-parameterized NN layer. We then harness the relationship between the theoretical properties and the minimum eigenvalue of the Pearson correlation matrix to craft our new proxies. We rigorously implement our proxies and extensively design the experiments. The experimental results show that MeCo and MeCo$_{opt}$ significantly outperforms the SOTA zero-cost proxies, which is also independent of the data and labels. Moreover, MeCo can be integrated into a complete NAS and enables us to efficiently find the architecture with the highest accuracy.

## Acknowledgment

We sincerely thank all the reviewers of NeurIPS'23 for their constructive suggestions which significantly improve the quality of our paper. The work was supported by National Key R&D Program of China [Grant No. 2022ZD0115901], in part by the National Natural Science Foundation of China (Grant No. 62177007), and the China-Central Eastern European Countries High Education Joint Education Project (Grant No. 202012).

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

# A  Detailed proofs

## A.1  Proof of Theorem 2

In this section, we present the proof for our main theorem (Theorem 2 in Section 3.4). We first introduce the following definitions and lemmas to better illustrate our proof.

**Lemma 1** (Lemma 3.1 in [31]). *If* $m = \Omega\left(\frac{n^2}{\lambda_0^2}\log\left(\frac{n}{\delta}\right)\right)$, *then with probability at least* $1 - \delta$, $\|\mathbf{H}(0) - \mathbf{H}^\infty\|_2 \leq \frac{\lambda_0}{4}$ *and* $\frac{3}{4}\lambda_0 \leq \lambda_{\min}(\mathbf{H}(0)) \leq \frac{5}{4}\lambda_0$.

**Definition 4.** *We denote the empirical indicator matrix as* $\mathbf{W}(0)$ *with entry* $[\mathbf{W}(0)]_{ij}$ *such that*

$$[\mathbf{W}(0)]_{ij} := \frac{1}{m}\sum_{r=1}^{m}\mathbb{I}\{\mathbf{w}_r^T(0)\mathbf{x}_i \geq 0, \mathbf{w}_r^T(0)\mathbf{x}_j \geq 0\}$$

*and indicator matrix as* $\bar{\mathbf{W}}$ *with entry* $[\bar{\mathbf{W}}]_{ij}$

$$[\bar{\mathbf{W}}]_{ij} := \mathbb{E}_{\mathbf{w}\sim N(\mathbf{0},\mathbf{I})}[\mathbb{I}\{\mathbf{w}^T\mathbf{x}_i \geq 0, \mathbf{w}^T\mathbf{x}_j \geq 0\}]$$

The indicator matrix $\bar{\mathbf{W}}$ has the following property.

**Proposition 1.** *If* $\mathbf{x}_i \nparallel \mathbf{x}_j$, *then* $[\bar{\mathbf{W}}]_{ij} > 0$.

**Proof.** If Proposition 1 does not hold, then $\exists i_0, j_0 \in [n] \times [n]$, satisfying

$$\begin{aligned}
0 =& [\bar{\mathbf{W}}]_{i_0 j_0} \\
=& \mathbb{E}_{\mathbf{w}\sim N(\mathbf{0},\mathbf{I})}[\mathbb{I}\{\mathbf{w}^T\mathbf{x}_{i_0} \geq 0, \mathbf{w}^T\mathbf{x}_{j_0} \geq 0\}] \\
=& P_{\mathbf{w}\sim N(\mathbf{0},\mathbf{I})}(\mathbf{w}^T\mathbf{x}_{i_0} \geq 0, \mathbf{w}^T\mathbf{x}_{j_0} \geq 0)
\end{aligned}$$

Thus $\mathbf{x}_{i_0} = -\mathbf{x}_{j_0}$, which contradicts the hypothesis $\mathbf{x}_{i_0} \nparallel \mathbf{x}_{j_0}$. ∎

In real-world datasets, the possibility of two different data being parallel is slight. Thus, proposition 1 holds in general. We further introduce Weyl inequality as follows:

**Lemma 2** (Weyl inequality [39]). *Let* $\mathbf{A}, \mathbf{B} \in \mathbb{R}^{n\times n}$ *be Hermitian matrices, and let the eigenvalues of* $\mathbf{A}$, $\mathbf{B}$, *and* $\mathbf{A}+\mathbf{B}$ *be* $\{\lambda_i(\mathbf{A})\}_{i=1}^n$, $\{\lambda_i(\mathbf{B})\}_{i=1}^n$ *and* $\{\lambda_i(\mathbf{A}+\mathbf{B})\}_{i=1}^n$, *respectively. The eigenvalues of each matrix are arranged in ascending order. Then we have*

$$\lambda_i(\mathbf{A} + \mathbf{B}) \leq \lambda_{i+j}(\mathbf{A}) + \lambda_{n-j}(\mathbf{B}), \quad j = 0, 1, \ldots, n - i \tag{14}$$

*for each* $i = 1, \ldots, n$, *with equality for some pair* $i, j$ *if and only if there is a nonzero vector* $\mathbf{x}$ *such that* $\mathbf{A}\mathbf{x} = \lambda_{i+j}(\mathbf{A})\mathbf{x}$, $\mathbf{B}\mathbf{x} = \lambda_{n-j}(\mathbf{B})\mathbf{x}$, *and* $(\mathbf{A} + \mathbf{B})\mathbf{x} = \lambda_i(\mathbf{A} + \mathbf{B})\mathbf{x}$. *Also,*

$$\lambda_{i-j+1}(\mathbf{A}) + \lambda_j(\mathbf{B}) \leq \lambda_i(\mathbf{A} + \mathbf{B}), \quad j = 1, \ldots, i \tag{15}$$

*for each* $i = 1, \ldots, n$, *with equality for some pair* $i, j$ *if and only if there is a nonzero vector* $\mathbf{x}$ *such that* $\mathbf{A}\mathbf{x} = \lambda_{i-j+1}(\mathbf{A})\mathbf{x}$, $\mathbf{B}\mathbf{x} = \lambda_j(\mathbf{B})\mathbf{x}$, *and* $(\mathbf{A} + \mathbf{B})\mathbf{x} = \lambda_i(\mathbf{A} + \mathbf{B})\mathbf{x}$. *If* $\mathbf{A}$ *and* $\mathbf{B}$ *have no common eigenvector, then inequality (14) and (15) are strict inequality.*

Now, we provide the full proof of Theorem 2.

**Theorem 2.** Suppose $f$ is an NN with a single hidden layer and ReLU activation function. Assume $\mathbf{X} \in \mathbb{R}^{d\times n}$, $\mathbf{w}(0) \sim N(\mathbf{0}, \mathbf{I})$, $P(\mathbf{X}) \succ 0$, $p_0 := \lambda_{\min}(P(\mathbf{X}))$, and hidden nodes $m = \Omega\left(\frac{n^6 d^2}{\lambda_0^4 \delta^3}\right)$, then the following formula holds with probability at least $1 - \delta$ over the initialization

$$\|f(\mathbf{W}(t), \mathbf{a}, \mathbf{X}) - \mathbf{y}\|_2^2 \leq \exp(-cp_0 t)\|f(\mathbf{W}(0), \mathbf{a}, \mathbf{X}) - \mathbf{y}\|_2^2$$

where $c$ is a constant depending on $m$ and $d$.

*Proof.* To simplify our proof, we assume $\mu_{\mathbf{x}} = 0$, $\sigma_{\mathbf{x}} = 1$, and $\|\mathbf{x}\|_2 \leq C$ for all $\mathbf{x} \in \mathbf{X}$. Recall that

$$[\mathbf{H}(0)]_{ij} = \mathbf{x}_i^T \mathbf{x}_j \frac{1}{m} \sum_{r=1}^m \mathbb{I}\{\mathbf{w}_r^T(0)\mathbf{x}_i \geq 0, \mathbf{w}_r^T(0)\mathbf{x}_j \geq 0\} = \mathbf{x}_i^T \mathbf{x}_j [\mathbf{W}(0)]_{ij}$$

Due to $\mathbb{I}\{\mathbf{w}_r^T(0)\mathbf{x}_i \geq 0, \mathbf{w}_r^T(0)\mathbf{x}_j \geq 0\}$ is an independent random variable between 0 and 1, then by Hoeffding's inequality [61], the following inequality holds with probability $1 - \delta$:

$$[\mathbf{W}(0)]_{ij} \geq [\bar{\mathbf{W}}]_{ij} - \frac{2\sqrt{\log(1/\delta)}}{\sqrt{m}}$$

Let $\mu_0 = \min_{(i,j) \in [n] \times [n]} [\bar{\mathbf{W}}]_{ij}$, and choose $m > \frac{16 \log(1/\delta)}{\mu_0^2}$, then we have

$$[\mathbf{W}(0)]_{ij} \geq \mu_0 - \frac{2\sqrt{\log(1/\delta)}}{\sqrt{m}} \geq \frac{2\sqrt{\log(1/\delta)}}{\sqrt{m}} =: c(m)$$

Define the matrix $\mathbf{M}$ with entry $[\mathbf{M}]_{ij}$ as

$$[\mathbf{M}]_{ij} := [\mathbf{H}(0)]_{ij} - c(m,d)[P(\mathbf{X})]_{ij}$$

where $c(m,d) = c(m)(d-1)$. We claim that if $m$ is large enough, then $\mathbf{M}$ is positive definite. To clarify this statement, we consider the gap between $\mathbf{H}(0)$ and $\mathbf{M}$:

$$\|\mathbf{H}(0) - \mathbf{M}\|_2 = \|c(m,d)P(\mathbf{X})\|_2 \leq c(m,d)\|P(X)\|_F \leq c(m,d)n^2 C^2$$

If we choose $m > \frac{64 \log(1/\delta)(d-1)^2 n^4 C^4}{\lambda_0^2}$, we have

$$\|\mathbf{H}(0) - \mathbf{M}\|_2 \leq \frac{\lambda_0}{4}$$

Then the following formula holds by matrix perturbation theory (Corollary 6.3.8 in [39])

$$0 < \frac{\lambda_0}{2} \leq \lambda_{\min}(\mathbf{H}(0)) - \frac{\lambda_0}{4} \leq \lambda_{\min}(\mathbf{M})$$

which indicates that $\mathbf{M}$ is positive definite. We next exert Lemma 2 on $\mathbf{M}$ and get

$$0 < \lambda_{\min}(\mathbf{H}(0) - c(m,d)P(\mathbf{X})) < \lambda_{\min}(\mathbf{H}(0)) - \lambda_{\min}(c(m,d)\mathbf{P}(\mathbf{X}))$$

That means

$$0 < c(m,d)p_0 \leq \lambda_{\min}(\mathbf{H}(0)) \leq \frac{5}{4}\lambda_0 \tag{16}$$

Therefore, combined with Theorem 1, we have

$$\begin{aligned}
&\|f(\mathbf{W}(t), \mathbf{a}, \mathbf{X}) - \mathbf{y}\|_2^2 \\
&\leq \exp(-\lambda_0 t)\|f(\mathbf{W}(0), \mathbf{a}, \mathbf{X}) - \mathbf{y}\|_2^2 \\
&\leq \exp(-c(m,d)p_0 t)\|f(\mathbf{W}(0), \mathbf{a}, \mathbf{X}) - \mathbf{y}\|_2^2
\end{aligned}$$

where $c(m,d)$ is a constant depending on $m$ and $d$. This completes the proof of **Theorem 2**. ∎

## A.2 Proof of Theorem 3

**Proof.** By Corollary 3.10 and Remark 3.11 in [40], we have

$$\mathbb{E}[L(W)] \le O\left(c \cdot \sqrt{\frac{\mathbf{y}^T(\mathbf{H}^\infty)^{-1}\mathbf{y}}{N}}\right) + O\left(\sqrt{\frac{\log(1/\delta)}{N}}\right)$$

According to the courant minimax principle[62], D.2 in [26], and inequality 16, we get

$$\mathbf{y}^T(\mathbf{H}^\infty)^{-1}\mathbf{y} \le \frac{\mathbf{y}^T\mathbf{y}}{\lambda_{\min}(\mathbf{H}^\infty)} \le c \cdot \frac{\mathbf{y}^T\mathbf{y}}{p_0}.$$

Thus, we have

$$\mathbb{E}[L(W)] \le O\left(c \cdot \sqrt{\frac{\mathbf{y}^T(\mathbf{H}^\infty)^{-1}\mathbf{y}}{N}}\right) + O\left(\sqrt{\frac{\log(1/\delta)}{N}}\right)$$
$$\le O\left(c \cdot \sqrt{\frac{\mathbf{y}^T\mathbf{y}}{Np_0}}\right) + O\left(\sqrt{\frac{\log(1/\delta)}{N}}\right)$$

∎

## B Zero-cost proxies

In this section, we provide the details of the baseline zero-cost proxies. Suppose $L$ is the loss function and $\theta$ is the parameters of an initialized network. We denote $\circ$ as the Hadamard product. The concrete formulations of the existing zero-cost proxies are as follows.

- snip. Lee et al. [20] use the changes in loss caused by the parameter perturbations to measure the importance of the parameters in an initialized network, such that

$$\mathcal{S}_{\mathsf{snip}}(\theta) = \left|\frac{\partial L}{\partial \theta} \circ \theta\right|$$

- grasp. Wang et al. [21] replace the loss change in snip with the change of the gradient norm to establish the proxy, such that

$$\mathcal{S}_{\mathsf{grasp}}(\theta) = -(H\frac{\partial L}{\partial \theta}) \circ \theta$$

where $H$ is the Hessian.

- synflow. To avoid layer collapse, Tanaka et al. [22] utilize the product of all parameters in the network during the parameter perturbation to represent the loss, such that

$$\mathcal{S}_{\mathsf{synflow}}(\theta) = \frac{\partial L}{\partial \theta} \circ \theta$$

- grad_norm. Abdelfattah et al. [8] adopt the $l_2$ norm of the gradients in an initialized work as a proxy, such that

$$\mathcal{S}_{\mathsf{grad\_norm}} = \left\|\frac{\partial L}{\partial \theta}\right\|_2$$

- jacov/NWOT. Mellor et al. [7] use the correlation of activations within a network as a proxy to evaluate the performance of the network, such that

$$\mathcal{S}_{\text{jacov}} = \log|\mathbf{K}_H|, \quad \mathbf{K}_H = \begin{pmatrix} N_A - d_H(\mathbf{c}_1, \mathbf{c}_1) & \cdots & N_A - d_H(\mathbf{c}_1, \mathbf{c}_N) \\ \vdots & \ddots & \vdots \\ N_A - d_H(\mathbf{c}_N, \mathbf{c}_1) & \cdots & N_A - d_H(\mathbf{c}_N, \mathbf{c}_N) \end{pmatrix}$$

where $N_A$ is the number of rectified linear units, $d_H(\mathbf{c}_i, \mathbf{c}_j)$ represents the Hamming distance between two binary codes $\mathbf{c}_i$ and $\mathbf{c}_j$.

- NTK. Chen et al. [24] propose to use the condition number of NTK to measure the trainability of the networks, such that

$$\kappa_{\mathcal{N}} = \frac{\lambda_{\max}(\hat{\Theta}_{train})}{\lambda_{\min}(\hat{\Theta}_{train})}$$

where $\hat{\Theta}_{train}$ stands for NTK of the networks. In our paper, we calculate the Spearman correlation coefficient between $1/\kappa_{\mathcal{N}}$ and the test accuracy of the networks.

- zen. Lin et al. [27] propose Zen-Score, in which they design an efficient zero-cost proxy with Gaussian random inputs, such that

$$\mathcal{S}_{\text{zen}} = \log\left(\mathbb{E}_{\mathbf{x}, \epsilon}\|f(\mathbf{x}; \theta) - f(\mathbf{x} + \alpha\epsilon; \theta)\|_F\right) + \sum_i \log\left(\sqrt{\sum_j \sigma_{i,j}^2/m}\right)$$

where $\sigma_{i,j}$ is the mini-batch standard deviation statistic of the $j$-th channel in BN.

- NASI. Shu et al. [52] propose NASI to evaluate the networks by approximating the trace of the NTK, such that

$$\text{NASI} = m\gamma^{-1} \left\| b^{-1} \sum_{x \in \mathcal{X}_j} \nabla_{\theta_0(\mathcal{A})} \mathcal{L}_x \right\|_2^2$$

where $\mathcal{X}_j$ is a mini-batch of data with size $|\mathcal{X}_j| = b$.

- KNAS. Xu et al. [51] propose to use gradient kernel to evaluate the networks, such that

$$g = \frac{1}{n^2} \sum_{i,j} \left(\frac{\partial y_j^L(t)}{\partial \mathbf{w}(t)}\right) \left(\frac{\partial y_i^L(t)}{\partial \mathbf{w}(t)}\right)^T$$

where $y^L$ is the output of $L$-th layer.

- GradSign. Zhang and Jia [53] propose GradSign, in which they analyze the sample-wise optimization landscape of the networks, such that

$$\text{GradSign} = \sum_k |\sum_i sign([\nabla_\theta l(f_\theta(x_i), y_i)|_{\theta_0}]_k)|$$

- ZiCo. Li et al. [9] explore the effect of gradient properties on network performance. They use absolute mean and standard deviation values of gradients to evaluate network performance:

$$\text{ZiCo} = \sum_{l=1}^{D} \log\left(\sum_{\omega \in \theta_l} \frac{\mathbb{E}[|\partial L(\mathbf{X}_i, \mathbf{y}_i; \theta)/\partial\omega|]}{\sqrt{Var(|\partial L(\mathbf{X}_i, \mathbf{y}_i; \theta)/\partial\omega|)}}\right), \quad i \in [N]$$

where $N$ and $D$ represent the number of batches and network layers, respectively. $\theta_l$ represents the parameters of the $l$-th layers.

## C   Algorithm of MeCo-based NAS

We slightly abuse the notation and denote MeCo as our proxy function for the network. We adopt the Zero-Cost-PT [55] to integrate MeCo to a zero-shot NAS. The algorithm of our MeCo-based NAS is summarized in Algorithm 1. We denote $A_0$ as an untrained supernet, $e_t$ represents the $t$-th edge, which stands for mixed operation in the search cells, and $e_{t,k}$ is the $k$-th operation of $t$-th edge. We denote $\mathcal{E}$, $\mathcal{N}$, and $\mathcal{O}$ as the set of edges, nodes, and candidate operations in the search cells, respectively. For any node $n \in \mathcal{N}$, we use $\mathcal{E}(n)$ as the set of its input edges, and $e_n^k$ is the $k$-th element of $\mathcal{E}(n)$. Note that we can use other zero-cost proxies as described in section B to replace MeCo in Algorithm 1.

---

**Algorithm 1** MeCo-based NAS

---

**Require:** $A_0$: An untrained supernet; $\mathcal{E}$: The set of edges in search cells; $\mathcal{N}$: The set of nodes in search cells; $\mathcal{O}$: The set of candidate operations; $N$: the number of candidate networks.
**Ensure:** The best network $A_{best}$.
 1: // Stage 1: Architecture Proposal
 2: $C = \emptyset$;
 3: **for** $i = 1; i \leq N; i + +$ **do**
 4:     **for** $j = 1; j \leq |\mathcal{E}|; j + +$ **do**
 5:         Randomly choose an un-discretized edge $e_t$
 6:         Choose the best edge from the supernet, s.t.

$$e_{t,best} = \underset{1 \leq k \leq |\mathcal{O}|}{\arg\min} \, \mathsf{MeCo}(A_0/e_{t,k})$$

 7:         Use operation $e_{t,best}$ to substitute $e_t$
 8:     **end for**
 9:     $A_{|\mathcal{E}|}$ consists of $\{e_{t,best}|1 \leq t \leq |\mathcal{E}|\}$
10:     **for** $j = 1; j < |\mathcal{N}|, j + +$ **do**         ▷ prune the edges of the obtained architecture $A_{|\mathcal{E}|}$
11:         Randomly select an unselected node $n \in \mathcal{N}$
12:         **for** $k = 1; k < |\mathcal{E}(n)|; k + +$ **do**
13:             Calculate MeCo of the architecture $A_{|\mathcal{E}|}/e_n^k$
14:         **end for**
15:         Retain edges $e_n^1, e_n^2$ with the 1st and 2nd best MeCo value, and remove the other edges
16:     **end for**
17:     Get the candidate networks $A_i$ that consist of $\{e_{t,best}|1 \leq t \leq \mathcal{E}\}$, and append it to the set $C$
18: **end for**
19: // Stage 2: Architecture Validation
20: Get the best network:

$$A_{best} = \underset{1 \leq i \leq N}{\arg\max} \, \mathsf{MeCo}(A_i), \; s.t. \; A_i \in C$$

---

## D   Experimental configurations

We use the same settings for the experiments as in [55]. We summarized the configurations of the searching phase and training phase on CIFAR-10 and CIFAR-100 in Table 6 and Table 7, respectively. For ZiCo, we use two mini-batch of data, which has a size of 64. For the other baseline proxies, we use one mini-batch of data. On the other hand, our MeCo only uses one random data $\mathbf{x} \in \mathbb{R}^{1 \times 3 \times 32 \times 32}$.

## E   More experimental results of $\rho$

### E.1   MeCo on NASBench-201 and NASBench-301 with three extra datasets

We evaluate MeCo and MeCo$_{opt}$ on NASBench-201 and NASBench-301 with Spherical-CIFAR-100, NinaPro, and SVHN, respectively. We only use one random data $x \in \mathbb{R}^{3 \times 32 \times 32}$ for MeCo and MeCo$_{opt}$. The results are summarized in Table 8. Our MeCo achieves competitive results on most benchmarks and achieves the best results on NASBench-201-SVHN, where the Spearman correlation coefficient is $\rho = 0.88$.

Table 6: The settings of Zero-Cost-PT with all proxies in DARTS-CNN for CIFAR-10

| Settings | Searching phase | | Training phase | |
|---|---|---|---|---|
| | Baselines | MeCo (Ours) | Baselines | MeCo (Ours) |
| batch size | 64 | 1 | 96 | 96 |
| cutout | True | False | True | True |
| cutout length | 16 | - | 16 | 16 |
| learning rate | 0.025 | 0.025 | 0.025 | 0.025 |
| learning rate min | 0.001 | 0.001 | - | - |
| momentum | 0.9 | 0.9 | 0.9 | 0.9 |
| weight decay | 3e-4 | 3e-4 | 3e-4 | 3e-4 |
| grad clip | 5 | 5 | 5 | 5 |
| init channels | 16 | 16 | 36 | 36 |
| layers | 8 | 8 | 20 | 20 |
| drop path prob | - | - | 0.2 | 0.2 |

Table 7: The settings of Zero-Cost-PT with all proxies in DARTS-CNN for CIFAR-100

| Settings | Searching phase | | Training phase | |
|---|---|---|---|---|
| | Baselines | MeCo (Ours) | Baselines | MeCo (Ours) |
| batch size | 64 | 1 | 96 | 96 |
| cutout | True | False | True | True |
| cutout length | 16 | - | 16 | 16 |
| learning rate | 0.025 | 0.025 | 0.025 | 0.025 |
| learning rate min | 0.001 | 0.001 | - | - |
| momentum | 0.9 | 0.9 | 0.9 | 0.9 |
| weight decay | 3e-4 | 3e-4 | 3e-4 | 3e-4 |
| grad clip | 5 | 5 | 5 | 5 |
| init channels | 16 | 16 | 16 | 16 |
| layers | 8 | 8 | 20 | 20 |
| drop path prob | - | - | 0.2 | 0.2 |

Table 8: Comparisons of $\rho$ with baselines using NASBench-301 and NASBench-201 on three extra datasets

| Method | NASBench-301 | | | NASBench-201 | | |
|---|---|---|---|---|---|---|
| | Sph-Cifar100 | NinaPro | SVHN | Sph-Cifar100 | NinaPro | SVHN |
| grasp | 0.13 | 0.04 | 0.18 | -0.01 | -0.01 | 0.62 |
| fisher | 0.00 | -0.11 | 0.05 | 0.07 | **-0.38** | 0.71 |
| grad_norm | -0.00 | **-0.20** | 0.42 | -0.08 | -0.23 | 0.77 |
| snip | -0.01 | -0.10 | 0.38 | -0.09 | -0.28 | 0.76 |
| synflow | 0.05 | -0.07 | 0.50 | 0.13 | 0.02 | 0.71 |
| l2_norm | **0.12** | -0.07 | **0.70** | -0.00 | 0.02 | 0.67 |
| #params | 0.07 | -0.07 | **0.70** | -0.14 | -0.11 | 0.72 |
| zen | 0.07 | -0.09 | 0.68 | 0.23 | 0.15 | 0.18 |
| jacov | 0.08 | 0.13 | -0.36 | **-0.41** | 0.29 | 0.67 |
| nwot | 0.05 | 0.02 | 0.64 | -0.02 | 0.06 | 0.76 |
| MeCo (**Ours**) | **-0.05** | **-0.11** | **0.68** | **-0.23** | **0.02** | **0.88** |
| MeCo$_{opt}$ (**Ours**) | **0.03** | **0.12** | **0.68** | - | - | - |

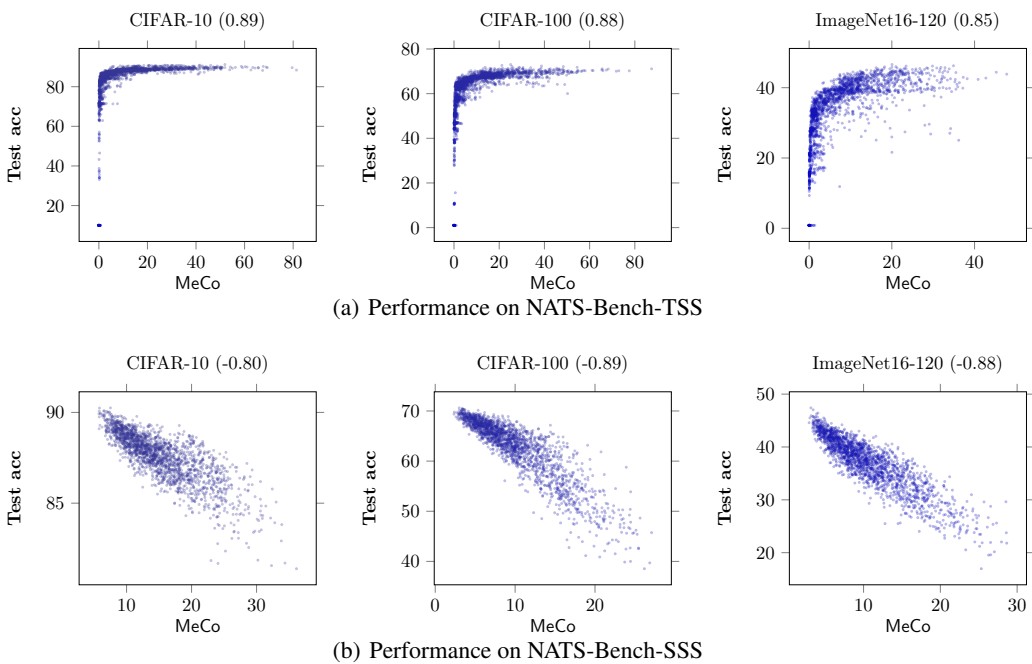

(a) Performance on NATS-Bench-TSS

(b) Performance on NATS-Bench-SSS

Figure 4: Relationship between our zero-cost proxies and test accuracy for NATS-Bench-TSS and NATS-Bench-SSS on three datasets.

## E.2    MeCo **on NAS-Bench-101**

We present the Spearman correlation coefficient between all zero-cost proxies and the test accuracy of the networks in NAS-Bench-101. Our MeCo uses one data of CIFAR-10, i.e., $\mathbf{x} \in \mathbb{R}^{1 \times 3 \times 32 \times 32}$ while the baselines adopt a batch of samples as input. The results are summarized in Table 9.

Table 9: Spearman correlation coefficients $\rho$ between zero-cost proxies and test accuracy on NAS-Bench-101

| Dataset | MeCo | Baselines | | | | | | | |
|---|---|---|---|---|---|---|---|---|---|
| | | grasp [21] | fisher[23] | grad_norm[8] | snip[20] | synflow [22] | jacov[7] | zen[27] | ZiCo [9] |
| CIFAR-10 | 0.44 | -0.33 | -0.27 | -0.32 | -0.25 | 0.36 | -0.35 | **0.63** | **0.63** |

MeCo achieves competitive results on NAS-Bench-101 with CIFAR-10, i.e., the Spearman correlation coefficient obtains 0.44, which is higher than the majority of the baselines. However, zen and ZiCo perform better than MeCo.

## E.3    MeCo **on TransBench-101**

We further evaluate our MeCo on diverse tasks. Specifically, we compare MeCo and the other proxies on Transbench-101-Micro and Transbench-101-Macro [47], respectively.

**TransBench-101-Micro.** We evaluate MeCo and MeCo_opt on TransBench-101-Micro with ten tasks. We calculate the Spearman correlation between our proxies and the accuracy of the networks. The results are summarized in Table 10. MeCo achieves the best performance on the three tasks, i.e., Class Objection, Spherical-Cifar100, and NinaPro, which are 0.58, 0.85, and 0.47, respectively. It can be seen that MeCo_opt effectively improves the performance of MeCo. For example, MeCo_opt improves MeCo from 0.62 to 0.77 on segmentation tasks.

**TransBench-101-Macro.** We compare MeCo, MeCo_opt and the baselines on TransBench-101-Macro with seven tasks. The results are summarized in Table 11. Our proxies achieve the best performance on Autoencoding, Jigsaw, and Surface Normal, which are 0.74, 0.48, and 0.80, respectively. Our proxies also achieve competitive results on the remaining tasks. Experimental results show that

our proxies can be used for diverse tasks. It can be demonstrated that MeCo and MeCo$_{opt}$ become ineffective on a few tasks, such as Room Layout. We would like to note that the existing proxies do not achieve a high correlation on all tasks consistently.

Table 10: Comparisons of $\rho$ with baselines using Transbench-101-Micro on Ten Tasks

| Approach | Autoencoding | Class Objection | Scene Classification | Jigsaw | Surface Normal | Segmantation | Room Layout | Spherical -Cifar100 | NinaPro | SVHN |
|---|---|---|---|---|---|---|---|---|---|---|
| grasp | -0.12 | -0.22 | -0.43 | -0.12 | 0.01 | 0.00 | -0.29 | -0.03 | -0.20 | -0.24 |
| fisher | **-0.58** | 0.44 | -0.13 | 0.30 | 0.16 | 0.12 | 0.30 | 0.72 | 0.42 | 0.81 |
| grad_norm | -0.32 | 0.39 | -0.33 | 0.36 | 0.36 | 0.60 | 0.25 | 0.72 | 0.40 | 0.78 |
| snip | -0.27 | 0.45 | -0.14 | 0.41 | 0.49 | 0.68 | 0.32 | 0.76 | 0.42 | 0.83 |
| synflow | 0.00 | 0.48 | 0.27 | 0.47 | 0.00 | 0.00 | 0.30 | 0.79 | 0.45 | **0.92** |
| l2_norm | 0.04 | 0.32 | 0.28 | 0.35 | 0.50 | 0.48 | 0.18 | 0.53 | 0.36 | 0.52 |
| #params | -0.01 | 0.45 | 0.32 | 0.44 | 0.62 | 0.68 | 0.30 | 0.79 | 0.36 | 0.76 |
| zen | 0.14 | 0.54 | 0.27 | 0.51 | 0.71 | 0.67 | 0.38 | 0.67 | 0.42 | 0.74 |
| jacov | 0.18 | 0.51 | 0.19 | **0.56** | **0.75** | **0.80** | **0.40** | 0.71 | 0.40 | 0.77 |
| nwot | 0.03 | 0.39 | **0.89** | 0.42 | 0.57 | 0.53 | 0.25 | 0.64 | 0.38 | 0.63 |
| zico | 0.35 | - | 0.71 | 0.52 | 0.68 | - | - | - | - | - |
| MeCo (Ours) | **0.03** | 0.58 | 0.62 | 0.45 | 0.65 | 0.62 | -0.25 | **0.85** | **0.47** | 0.88 |
| MeCo$_{opt}$ (Ours) | **0.03** | **0.59** | 0.64 | **0.47** | 0.67 | 0.77 | 0.26 | **0.85** | **0.47** | 0.88 |

Table 11: Comparisons of $\rho$ with baselines using Transbench-101-Macro on Seven Tasks

| Approach | Autoencoding | Class Objection | Scene Classification | Jigsaw | Surface Normal | Segmantation | Room Layout |
|---|---|---|---|---|---|---|---|
| grasp | -0.02 | -0.64 | -0.43 | -0.26 | -0.05 | -0.02 | -0.26 |
| fisher | -0.19 | -0.30 | -0.13 | -0.26 | 0.15 | 0.03 | -0.26 |
| grad_norm | 0.31 | -0.56 | -0.33 | -0.27 | 0.35 | 0.21 | -0.27 |
| snip | 0.20 | -0.38 | -0.14 | -0.19 | 0.45 | 0.27 | -0.19 |
| synflow | 0.00 | 0.12 | 0.27 | 0.34 | 0.00 | 0.00 | 0.34 |
| l2_norm | -0.20 | 0.08 | 0.28 | 0.15 | 0.30 | 0.18 | 0.15 |
| #params | -0.18 | 0.16 | 0.32 | 0.15 | 0.30 | 0.06 | 0.15 |
| zen | -0.01 | 0.10 | 0.27 | 0.24 | 0.38 | 0.27 | 0.24 |
| jacov | 0.45 | 0.07 | 0.19 | 0.19 | 0.50 | 0.57 | 0.19 |
| nwot | **0.67** | **0.83** | **0.89** | **0.48** | 0.78 | **0.80** | **0.76** |
| MeCo (Ours) | 0.51 | 0.59 | 0.81 | 0.17 | **0.80** | 0.62 | 0.23 |
| MeCo$_{opt}$ (Ours) | 0.74 | 0.73 | 0.76 | 0.48 | 0.76 | 0.63 | 0.33 |

### E.4  MeCo **on AutoFormer and MobileNet OFA**

**AutoFormer.**  Chen et al. [49] proposed a novel one-shot architecture search framework for transformer-based models. We load the trained supernets and regenerate the candidate subnets. We then re-evaluate the subnets to obtain the accuracy on ImageNet and compute MeCo. The correlation of MeCo and model accuracy on AutoFormer is 0.45.

**OFA.** To solve the problem of efficient inference across devices and resource constraints, Cai et al. [50] proposed to train a once-for-all (OFA) network, which supports diverse architectural settings. We randomly sample 1,000 subnets from the OFA network and use the accuracy of the predictor predictions as the test accuracy. The Spearman correlation between MeCo and test accuracy is 0.86.

## F    More experimental results of NAS with MeCo

In this section, we provide more results and comparisons of our zero-shot NAS on NATS-Bench-TSS and DARTS-CNN. In the following descriptions, we denote multi-shot, one-shot, and zero-shot as MS, OS, and ZS, respectively.

### F.1    Results on NATS-Bench-TSS

In the NATS-Bench-TSS search space, we evaluate all networks using zero-cost proxies and choose the Top-10 networks with the highest scores. Then we calculate the average test accuracy and standard deviation. The results are summarized in Table 12. As shown in the table, the networks searched by MeCo have the highest average precision on CIFAR-10, which is 0.08 higher than the best baseline proxy ZiCo. For CIFAR-100 and ImageNet16-120, MeCo achieves competitive results,

e.g., $70.86\%\pm0.96\%$ with CIFAR-100 and $42.59\%\pm1.77\%$ with ImageNet16-120. In all, the results demonstrate that our MeCo has a great advantage in evaluating network performance, considering MeCo only requires one data sample as input.

We further combine MeCo with Zero-Cost-PT [55] and search for the best architecture three times with different seeds. The accuracy of the selected architectures as well as the comparisons with the baseline methods are presented in Table 13. It can be shown from the results that our MeCo-based NAS achieves competitive results with the SOTA baselines, e.g., synflow, zen, and ZiCo. However, MeCo-based NAS invokes one data for a single forward pass, thus being more resource-saving.

Table 12: The average test accuracy of Top-10 architectures obtained by various zero-cost proxies on NATS-Bench-TSS using CIAFR-10, CIAFR-100, and ImageNet16-120, respectively

| Dataset | MeCo | Baselines | | | | | | | | |
| --- | --- | --- | --- | --- | --- | --- | --- | --- | --- | --- |
| | | grasp[21] | fisher [23] | grad_norm[8] | snip [20] | synflow[22] | jacov[7] | NTK[24] | zen[27] | ZiCo[9] |
| **CIFAR-10** | **93.64** ±**0.31** | 89.34 ±2.16 | 89.27 ±2.10 | 89.27 ±2.10 | 89.27 ±2.10 | 93.27 ±0.74 | 91.23 ±1.01 | 92.67 ±0.46 | 58.99 ±3.44 | 93.56 ±0.23 |
| **CIFAR-100** | 70.86 ±0.96 | 60.89 ±3.88 | 61.06 ±4.02 | 60.89 ±3.88 | 60.89 ±3.88 | **71.12** ±**1.59** | 68.50 ±1.21 | 69.31 ±1.17 | 12.73 ±1.33 | 70.64 ±0.28 |
| **ImageNet16-120** | 42.59 ±1.77 | 22.99 ±11.01 | 24.10 ±11.36 | 23.35 ±11.44 | 23.35 ±11.44 | 42.65 ±3.59 | 40.59 ±1.86 | 39.98 ±1.73 | 15.10 ±0.51 | **42.74** ±**1.78** |

Table 13: The test accuracy of optimal architectures obtained by Zero-Cost-PT with various zero-cost proxies on NATS-Bench-TSS using CIAFR-10, CIFAR-100, and ImageNet16-120, respectively

| Dataset | MeCo | Baselines | | | | | | | | |
| --- | --- | --- | --- | --- | --- | --- | --- | --- | --- | --- |
| | | grasp[21] | fisher [23] | grad_norm[8] | snip [20] | synflow[22] | jacov[7] | NTK[24] | zen[27] | ZiCo[9] |
| **CIFAR-10** | **93.76** ±**0** | 92.59 ±1.12 | 88.39 ±2.55 | 91.64 ±0.68 | 90.11 ±2.85 | **93.76** ±**0** | 91.92 ±2.40 | 92.61 ±0.65 | **93.76** ±**0** | **93.76** ±**0** |
| **CIFAR-100** | **71.11** ±**0** | 68.98 ±2.69 | 65.77 ±0.93 | 65.20 ±0.56 | 65.29 ±0.96 | **71.11** ±**0** | 69.67 ±2.39 | 68.27 ±2.34 | **71.11** ±**0** | **71.11** ±**0** |
| **ImageNet16-120** | **41.44** ±**0** | 35.29 ±8.03 | 28.91 ±5.18 | 35.82 ±3.99 | 37.38 ±4.41 | **41.44** ±**0** | 40.35 ±6.56 | 41.25 ±2.37 | **41.44** ±**0** | **41.44** ±**0** |

## F.2   Results on DARTS-CNN

For DARTS-CNN search space, we search the architectures by Zero-Cost-PT with different zero-cost proxies on CIFAR-100. Each searched network is trained five times, and the results are summarized in Table 14. There are two settings in our experiments on DARTS-CNN: networks initialized with 16 channels and trained as in Table 7, and networks initialized with 36 channels and trained as in [63]. It can be shown from Table 14 that MeCo achieves competitive results compared with MS, OS, and ZS baselines. More concretely, MeCo-based NAS obtains $19.33\%$ test error with 0.08 GPU days, which outperforms all the ZS methods under the same settings. On the other hand, compared with the manual, MS, and OS methods, MeCo is also competitive. For example, MeCo-based NAS achieves $83.14\%$ accuracy, which is only $0.34\%$ lower than the baseline method $\beta$-DARTS [64], but five times more efficient in computation.

We further visualize the networks searched by Zero-Cost-PT with different proxies on DARTS-CNN. The results on CIFAR-10 and CIFAR-100 are presented in Figure 6 and Figure 7, respectively.

## F.3   Results on MobileNet V3

We search the architectures on MobileNet space using Algorithm 1. We retrain the searched network using ImageNet-1K for 480 epochs with a batch size of 256 and input resolution $224 \times 224$. The results are summarized in Table 15. MeCo achieves $77.8\%$ Top-1 accuracy, which is $0.7\%$ higher than the SOTA baseline method ZiCo. Moreover, our method only takes 0.04 GPU days, which outperforms the SOTA methods.

# G   Limitations and future works

Table 14: Comparison of our method with SOTA NAS methods using DARTS-CNN and CIFAR-100.

| Approach | Test Error (%) | Search Cost (GPU Days) | Params (MB) | Method |
|---|---|---|---|---|
| DenseNet-BC[65] | 17.18 | - | 25.6 | Manual |
| NASNet-A[12] | 16.82 | 2000 | 3.3 | MS |
| DARTS(1st)[6] | 17.76 | 1.5 | 3.3 | OS |
| SNAS [66] | 17.55 | 1.5 | 2.8 | OS |
| P-DARTS[64] | $15.92 \pm 0.18$ | 0.4 | 3.7 | OS |
| R-DARTS[67] | $18.01 \pm 0.26$ | - | - | OS |
| PC-DARTS[63] | 16.9 | 0.1 | 3.6 | OS |
| $\beta$-DARTS[68] | $16.52 \pm 0.03$ | 0.4 | $3.83 \pm 0.08$ | OS |
| Zero-Cost-PT$_{\text{synflow}}^{\dagger}$ [22] | $19.82 \pm 0.35$ | 0.04 | 1.2 | ZS |
| Zero-Cost-PT$_{\text{fisher}}^{\dagger}$[23] | $21.14 \pm 0.24$ | 0.06 | 0.7 | ZS |
| Zero-Cost-PT$_{\text{grasp}}^{\dagger}$ [21] | $22.65 \pm 0.30$ | 0.13 | 0.7 | ZS |
| Zero-Cost-PT$_{\text{jacov}}^{\dagger}$[7] | $22.90 \pm 0.35$ | 0.04 | 0.6 | ZS |
| Zero-Cost-PT$_{\text{snip}}^{\dagger}$[20] | $19.95 \pm 0.28$ | 0.04 | 0.8 | ZS |
| Zero-Cost-PT$_{\text{NTK}}^{\dagger}$[24] | $20.30 \pm 0.33$ | 0.19 | 0.9 | ZS |
| Zero-Cost-PT$_{\text{ZiCo}\dagger}$[9] | $19.54 \pm 0.28$ | 0.06 | 1.1 | ZS |
| **Zero-Cost-PT$_{\text{MeCo}}^{\dagger}$** | $\mathbf{19.33 \pm 0.25}$ | 0.08 | 0.8 | ZS |
| **Zero-Cost-PT$_{\text{MeCo}}^{\ddagger}$** | $\mathbf{16.86 \pm 0.30}$ | 0.08 | 3.7 | ZS |

†: networks initialized with 16 channels and trained as settings in 7.

‡: networks initialized with 36 channels and trained as settings in [63].

Table 15: Comparison of our method with SOTA NAS methods using MobileNet and ImageNet-1K.

| Approach | Top-1 (%) | Search Cost (GPU Days) | Params (M) | Method |
|---|---|---|---|---|
| MobileNet-V3(1.0) | 75.2 | 288 | 5.3 | MS |
| PNAS | 74.2 | 224 | 5.1 | MS |
| DARTS | 73.3 | 4 | 4.7 | OS |
| PC-DARTS | 75.8 | 3.8 | - | OS |
| SPOS | 74.7 | 8.3 | - | OS |
| GreedyNAS | 74.9 | 7.6 | 3.8 | OS |
| TE-NAS | 75.5 | 0.17 | 5.4 | ZS |
| ZiCo | 77.1 | 0.4 | - | ZS |
| Zero-Cost-PT | 76.4 | 0.04 | 8.0 | ZS |
| Zen-score | 76.1 | 0.5 | - | ZS |
| **MeCo (Ours)** | **77.8** | **0.08** | 7.9 | ZS |

Finally, we demonstrate the limitations of this work and the possible directions of our future work. In Section 5, We propose an optimization method to alleviate the channel-sensitive issue of MeCo. This weight-sampling approach can be further improved in future work. Moreover, we would like to point out that the evaluation of a zero-cost proxy is often tied to the availability of benchmarks. Hench though the proxies are "zero cost", the evaluation of the proxies is strongly dependent on a benchmark, which in the first place is very expensive to create. Finally, although MeCo achieves the highest correlation with the test accuracy on multiple benchmarks, it shows near zero correlations on a few tasks (e.g., TransBench-101-Micro with Autoencoding). We will leave these issues as our future work.

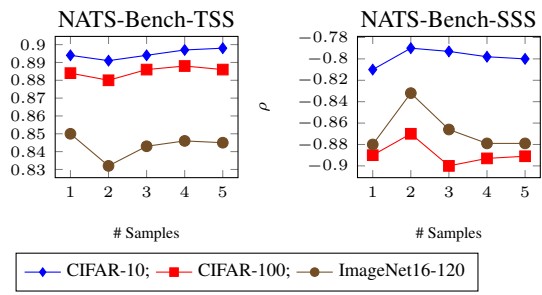

Figure 5: Spearman correlation coefficients of MeCo with different number of samples on NATS-Bench-TSS and NATS-Bench-SSS, respectively.

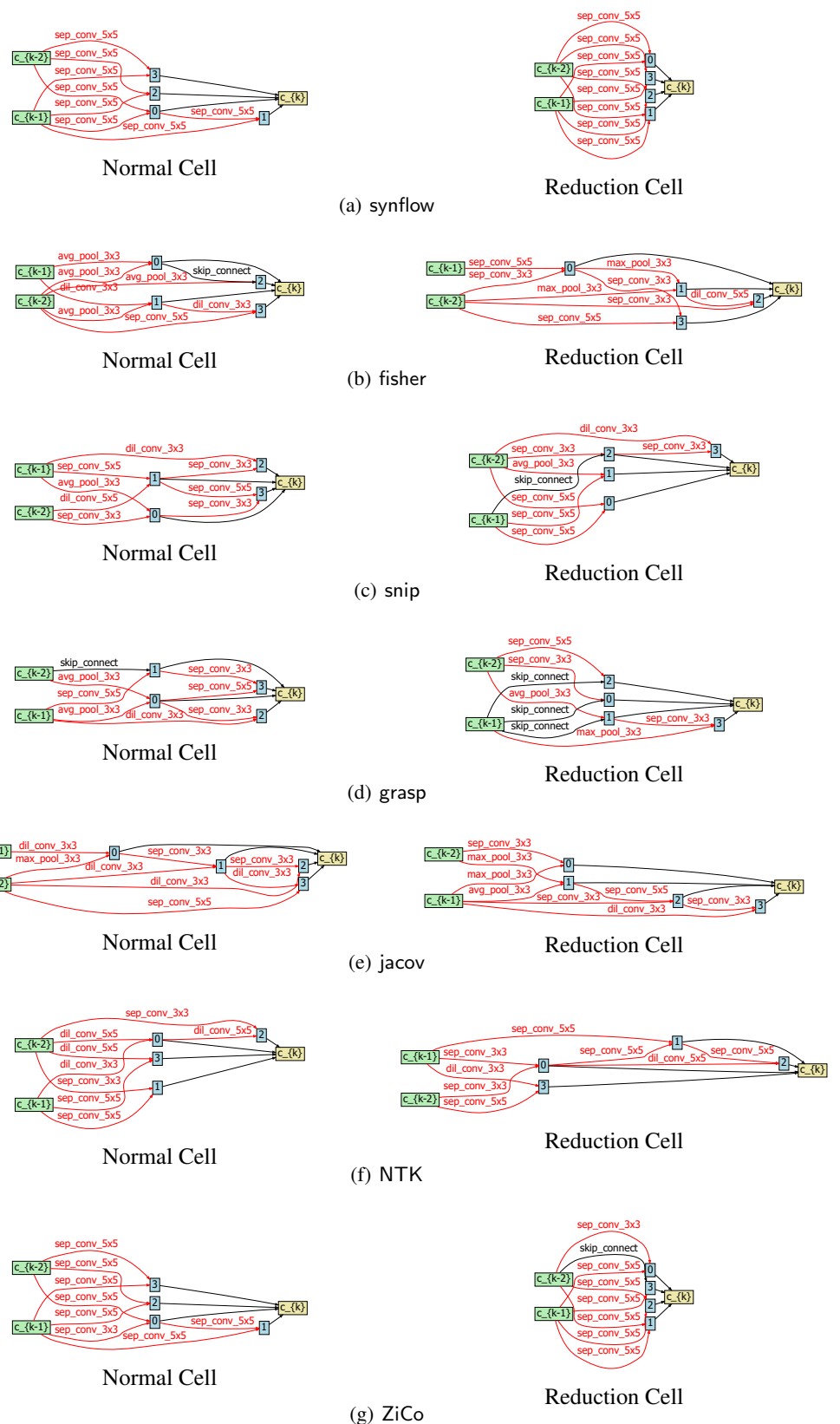

(a) synflow

(b) fisher

(c) snip

(d) grasp

(e) jacov

(f) NTK

(g) ZiCo

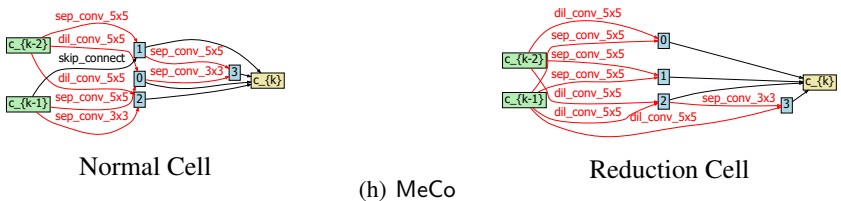

(h) MeCo

Figure 6: Cells found by Zero-Cost-PT with all zero-cost proxies on the DARTS-CNN search space using CIFAR-10

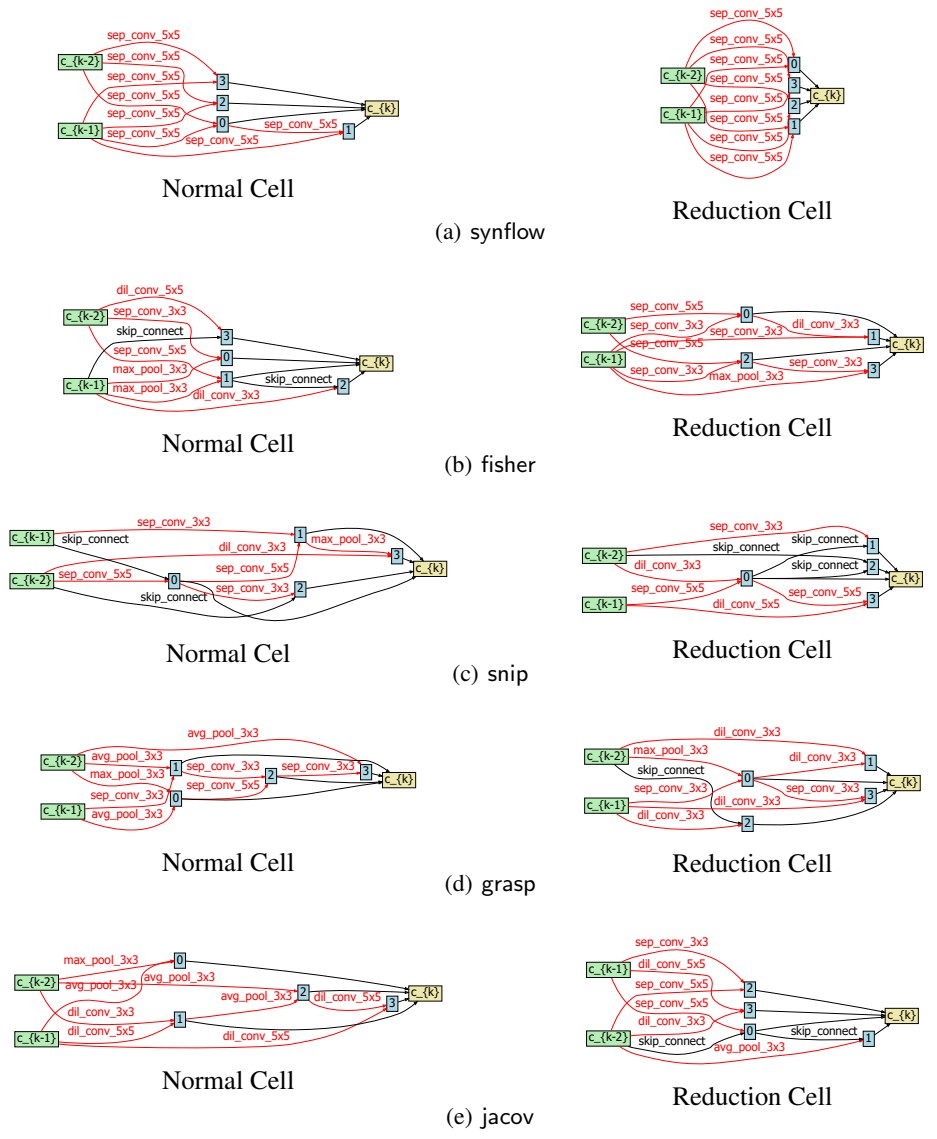

(a) synflow

(b) fisher

(c) snip

(d) grasp

(e) jacov

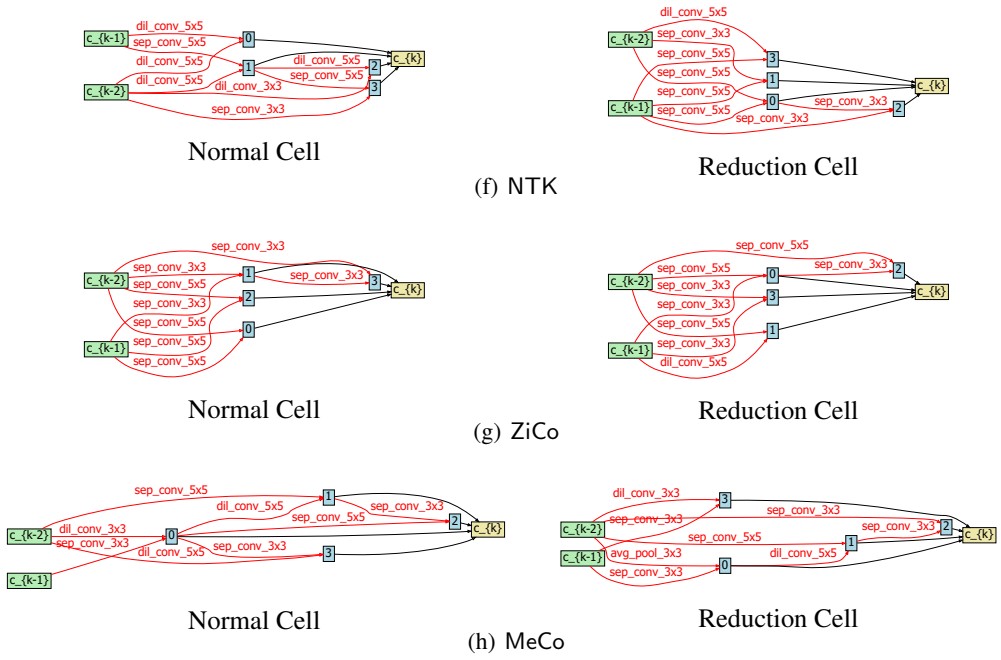

Figure 7: Cells found by Zero-Cost-PT with all zero-cost proxies on the DARTS-CNN search space using CIFAR-100

