| **CIFAR-100** | 70.86 $\pm$0.96 | 60.89 $\pm$3.88 | 61.06 $\pm$4.02 | 60.89 $\pm$3.88 | 60.89 $\pm$3.88 | **71.12** $\pm$**1.59** | 68.50 $\pm$1.21 | 69.31 $\pm$1.17 | 12.73 $\pm$1.33 | 70.64 $\pm$0.28 |
| **ImageNet16-120** | 42.59 $\pm$1.77 | 22.99 $\pm$11.01 | 24.10 $\pm$11.36 | 23.35 $\pm$11.44 | 23.35 $\pm$11.44 | 42.65 $\pm$3.59 | 40.59 $\pm$1.86 | 39.98 $\pm$1.73 | 15.10 $\pm$0.51 | **42.74** $\pm$**1.78** |

Table 8: The test accuracy of optimal architectures obtained by Zero-Cost-PT with various zero-cost proxies on NATS-Bench-TSS using CIAFR-10, CIFAR-100, and ImageNet16-120, respectively

| Dataset | MeCo | Baselines | | | | | | | | |
| | | grasp[8] | fisher [19] | grad_norm[10] | snip [7] | synflow[9] | jacov[11] | NTK[12] | zen[13] | ZiCo[17] |
|---|---|---|---|---|---|---|---|---|---|---|
| **CIFAR-10** | **93.76** $\pm$**0** | 92.59 $\pm$1.12 | 88.39 $\pm$2.55 | 91.64 $\pm$0.68 | 90.11 $\pm$2.85 | **93.76** $\pm$**0** | 91.92 $\pm$2.40 | 92.61 $\pm$0.65 | **93.76** $\pm$**0** | **93.76** $\pm$**0** |
| **CIFAR-100** | **71.11** $\pm$**0** | 68.98 $\pm$2.69 | 65.77 $\pm$0.93 | 65.20 $\pm$0.56 | 65.29 $\pm$0.96 | **71.11** $\pm$**0** | 69.67 $\pm$2.39 | 68.27 $\pm$2.34 | **71.11** $\pm$**0** | **71.11** $\pm$**0** |
| **ImageNet16-120** | **41.44** $\pm$**0** | 35.29 $\pm$8.03 | 28.91 $\pm$5.18 | 35.82 $\pm$3.99 | 37.38 $\pm$4.41 | **41.44** $\pm$**0** | 40.35 $\pm$6.56 | 41.25 $\pm$2.37 | **41.44** $\pm$**0** | **41.44** $\pm$**0** |

## F.2 Results on DARTS-CNN