# OpenReview forum: "MeCo: Zero-Shot NAS with One Data and Single Forward Pass via Minimum Eigenvalue of Correlation"
_NeurIPS.cc/2023/Conference — NeurIPS 2023 spotlight_

### Official Review · Reviewer_1Ted · 2023-07-05

**Soundness:** 3 good
**Presentation:** 3 good
**Contribution:** 3 good
**Rating:** 5
**Confidence:** 4

**Summary:**

This paper proposes a novel zero-cost proxy called MeCo. The proxy is data-independent, which requires only one random data with a single forward pass. It analyzed that the minimum eigenvalue of P(X) is correlated to the training convergence rate of the network. The experiments on several NAS benchmarks evaluate the effectiveness of MeCo.

**Strengths:**

1. The paper proposes a novel zero-cost proxy MeCo, which is theoretically guaranteed.
2. The ranking correlation results demonstrate the superiority of MeCo compared with other proxies.
3. The writing is good and easy to follow.

**Weaknesses:**

1. On Table 5, the performance gain of Zero-Cost-PT_MeCo is small compared to other Zero-Cost-PT methods. Besides,  the performance gain of Zero-Cost-PT_MeCo is lower than TE-NAS.
2. More experimental results should be provided to demonstrate the effectiveness of the proposed method, such as the ranking results on NAS-Bench-301 and the search results on ImageNet in DARTS search space.

**Questions:**

1. In the theoretical proof section, the paper mainly focuses on the network convergence rate and convolutional layer. I wonder how can the theory be connected with network generalization? And can the MeCo metric be applicable to the self-attention layer?
2. The paper proposes to use only one random data. But the ablation experiments in Figure 4 show that the optimal result is obtained when the data size is not equal to 1. How to explain this?
3. In Table 5, why is the search cost higher than the other gradient-based zero-cost methods? For example, Zero-Cost-PT_MeCo v.s. Zero-Cost-PT_ZiCo and Zero-Cost-PT_Snip, because the paper requires only one random data with a single forward pass which should be very low search cost.
4. The results on ImageNet in DARTS-CNN search space should be provided.
5. To show the effectiveness of MeCo, more ranking consistency experiments should be conducted in different search spaces, such as NAS-Bench-301.
6. Minor Error. In line 220, "0.6 to 0.9" -> "0.06 to 0.09".

**Limitations:**

See the Weaknesses and Questions above.

---

> ### Author Rebuttal · Authors · 2023-08-08
>
> Thanks a lot for your time and valuable comments and suggestions! We would like to respond to your reviews point by point.
>
> **Q1: How can the theory be connected with network generalization? Can the MeCo metric be applicable to the self-attention layer?**
>
> Answer: 1) MeCo can also reflect the generalization capacity of a network. We present the following theorem.
>
> **Theorem 3.** For an over-parameterized neural network, we denote the loss as $L(W)$. Let $y=(y_1, ..., y_N)^T$ ,  $\lambda_0=\lambda_{min}(H^{\infty})$. Let $\gamma$ be the step of SGD, $\gamma = \kappa C_1\sqrt{y^T(H^\infty)^{-1}y}/(m\sqrt N)$ for some small enough absolute constant $\kappa$. Under the assumption of Theorem 1,  for any $\delta\in(0,e^{-1}])$, there exists $m^\ast(\delta,N,\lambda_0)$, such that if $m\geq m^*$, then with probability at least $1-\delta$, we have
> $$
> E[L(W)] \leq O(C'\sqrt{\frac{y^T y}{p_0 N}})+ O(\sqrt{\frac{\log(1/\delta)}{N}})
> $$
> where $C, C', \delta$ are constants.
> The proof can be obtained from Corollary 3.10 of [1] and Section D.2 of [2]. Theorem 3 indicates that the generalization capacity is bounded by $p_0$. The expectation of the loss decreases when $p_0$ gets larger. This relationship is consistent with Theorem 2.
>
> We would like to point out that Theorem 3 indicates that the generalization capacity of a network is bounded by $p_0$. The expectation of the loss on the testing dataset decreases when $p_0$ gets larger. This relationship is also consistent with the impact of $p_0$ on the convergence rate.
>
> 2)Intuitively, MeCo can be applied to any architecture so long as the feature maps during the forward pass can be extracted. To test this, we adopt MeCo on a transformer benchmark (Autoformer). The correlation between MeCo and the model performance is 0.45, which proves the effectiveness of MeCo to an extent. The theory of MeCo is built on the over-parameterized NN layer, which is not the same as the self-attention layer. We will leave the extension of the theory part as our future work.
>
>
> **Q2: Unclear ablation study:**
>
> Answer: Figure 4 demonstrates MeCo v.s. the SIZE of the data sample, not the number of samples. It might be more appropriate to use the word "dimension" to avoid possible confusion. All the experiments on MeCo are conducted using one data sample, which would be first resized to a specific dimension. Figure 4 shows the best data dimension for SSS and TSS. We will clarify this in this section to avoid possible confusion.
>
> **Q3: In Table 5, why is the search cost higher than the other methods?**
>
> Answer: MeCo is more resource-saving because it only requires one forward pass. In Table 5, the main computation bottleneck is computing the minimum eigenvalue of the Pearson correlation matrix. For now, we use one CPU kernel without any thread-level parallelization or optimizations. We believe the time cost can be further reduced by adopting parallelization techniques.
>
> **Q4: The results on ImageNet:**
>
> Answer: Thanks for pointing out this issue. We agree that we should provide more results, especially on large-scale datasets. Unfortunately, it is relatively difficult to fully train the searched model in DARTS-CNN on ImageNet-1k in this short rebuttal period. So alternatively, we provide the results on ImageNet-1k with MobileNet v2-like search space. The results are shown in Table 4 in the attached PDF. We trained the searched optimal network with six RTX 3090 for about 130 hours. The Top-1 accuracy is **77.3\%**, which is higher than all the baseline approaches.
>
> **Q5: More results on other benchmarks.**
>
> Answer: We conducted more experiments on NASBench-301, NASBench-201, TransBench-101 (both Micro and Macro), MobileNet-like search space, etc. The experimental results can be found in Table 1, 2, and 3 in the attached PDF.
>
> For NAS-Bench-301, we utilize three extra datasets except for the standard CIFAR-10 following the settings in NASBench-zero-suite. The results show that MeCo achieves the highest correlation on CIFAR-10 among all the baselines (e.g., 0.04 higher than Zico). MeCo also provides competitive performance on NinaPro and SVHN (e.g., 0.09 and 0.02 lower than the best method). On Spherical-CIFAR-100, MeCo has an average correlation that is 0.08 lower than grasp. Yet MeCo outperforms grasp on all the remaining tasks. MeCo also achieves SOTA results on other benchmarks, please refer to the Tables in the attached PDF for more details.
>
> **Q6: In line 220, "0.6 to 0.9" -> "0.06 to 0.09".**
>
> Answer: Thanks for pointing this out! We will fix this typo.
>
> **Q7 (Other concerns in the *Weakness* part):The performance gain of Zero-Cost-PT_MeCo is lower than TE-NAS.**
>
> Answer: We acknowledge that MeCo achieves marginal performance gain using Zero-Cost-PT and lower accuracy than TE-NAS. We would like to point out that MeCo is designed as an evaluation metric of the architecture, which achieves SOTA correlation on TSS, SSS, NASBench-301 (added in rebuttal), and some tasks on Transbench-101 (added in rebuttal). These results reflect the capability of MeCo and demonstrate the possibility of a resource-saving and label-independent zero-cost proxy. On the other hand, almost all the zero-cost proxies need to be integrated with an external generation strategy to form a complete NAS (e.g., Zero-cost-PT). Thus, the performance of the proxy-based NAS relies on the generation strategy. Zero-cost-PT performs unconstrained NAS, which is different from the baselines, thus should not be interpreted as objectively better (p.9 in Zero-cost-PT). We will leave the research on the more appropriate generation algorithm as our future work.
>
> [1] Cao Y, Gu Q. Generalization bounds of stochastic gradient descent for wide and deep neural networks[J]. Advances in neural information processing systems, 2019, 32.
>
> [2] Zhu Z, Liu F, Chrysos G, et al. Generalization properties of NAS under activation and skip connection search[J]. Advances in Neural Information Processing Systems, 2022, 35: 23551-23565.

---

> ### Author Response · Authors · 2023-08-21
> **Sincerely expecting further discussions with reviewer 1Ted**
>
> Dear Reviewer 1Ted,
>
> Thanks for your constructive suggestions in your review. Since the discussion period is ending soon, have our previous responses addressed your concerns?
>
> We would be more than happy to further discuss with you and provide more information.
>
> Best, Authors, Paper 7200

---

> ### Comment · Reviewer_1Ted · 2023-08-21
> **Response to the Rebuttal**
>
> I have read all the responses. I would like to keep my score as borderline accept.

---

> > ### Author Response · Authors · 2023-08-21
> > **Thanks for your valuable comments!**
> >
> > Dear Reviewer 1Ted,
> >
> > Thanks for your valuable comments and suggestions!
> >
> > Best,
> >
> > Authors 7200

---

### Official Review · Reviewer_4Zmk · 2023-07-06

**Soundness:** 2 fair
**Presentation:** 3 good
**Contribution:** 2 fair
**Rating:** 7
**Confidence:** 5

**Summary:**

This paper proposes MeCo, a new zero-shot NAS method that can rank model architectures without training, without any backward pass, and just using 1 data sample (instead of a batch of data like existing methods). The method is evaluated on multiple NAS Benches and the authors have also run NAS experiments on CIFAR-10 and CIFAR-100 datasets. Rest of the NAS runs are done directly on NAS-Benches. The authors have also presented a simple theoretical result relating Pearson’s correlation to gram matrix which is sometimes used to prove effectiveness of zero-cost proxies.

**Strengths:**

The paper has following strengths:

1. The main strength of this idea is that it uses only one data sample and no backward pass. So, the proposed zero-shot method is quick.

2. On NATS-Bench-SSS and NATS-Bench-TSS, the proposed method achieves higher correlation than existing methods.

3. The paper is very well-written.

**Weaknesses:**

The paper has following weaknesses:

1. No results on zero-shot NAS runs on generic search spaces on ImageNet. Due to low search cost, many new proxies like ZiCo, ZenNAS, etc., have shown results on ImageNet and often beat multi-shot and one-shot methods by a large margin (in search time as well as accuracy). The authors should conduct this experiment with MeCo and present detailed results and comparisons to existing methods. In the current state, the total accuracy improvements compared to existing baselines like Zen, ZiCo, etc., are not too significant (e.g., see tables 4, 5, 6 in supplementary and table 4 in main paper). Significant improvements on ImageNet can make the paper's claims much more convincing.

2. Between NATS-Bench-TSS to NATS-Bench-SSS, the correlation changes from highly positive correlation to highly negative correlation (see Tables 1 and 2 in main paper). Authors show some results stating that this is related to large channel widths of networks which impacts the minimum eigenvalue. However, this may also mean there may be some channel widths where the MeCo correlations may become close to zero (since it changed from highly positive to highly negative when changing channel widths or other architecture properties). In case that happens, it may be problematic since zero correlations cannot really say anything about model quality. Can the authors conduct a detailed (theoretical/empirical) analysis towards this important issue? Under what conditions would MeCo work optimally?

3. It was not clear what kind of data size the authors changed in Section 5 ablation study. Did they change input feature dimensions or number of samples? If they did not ablate over number of samples used to calculate MeCo, they should conduct such a study (e.g., how does correlation vary if you use 16 samples instead of 1 sample, etc.).

4. Also, if they use only one data sample to calculate MeCo, what is the variance of MeCo across multiple samples from same or different classes?

5. Few other NAS-Benches are available and has been utilized in literature (e.g., TransNASBench and others used in ZiCo paper). More results can be shown on those benchmarks.

6. A couple other papers authors should also discuss in related work: (1) “Layerwise Dynamical Isometry” [ICLR 2020, https://arxiv.org/abs/1906.06307] that talks about signal propagation properties via layer Jacobians to improve training convergence of highly sparse networks. (2) NN-Mass [CVPR 2021, https://arxiv.org/abs/1910.00780] which explicitly relates topological properties of deep networks to gradient flow. NN-Mass does not even require any forward pass and ranks networks purely using their architecture's structural properties (although it works in very limited settings). In fact, these two works form the earliest precursors to the zero-shot NAS domain (and came before all other training free metrics) and must be discussed.


==== UPDATE AFTER REBUTTAL ====
I have read the author response and other reviews. The new data and analyses provided by the authors have improved the paper compared to the initial submission. However, I still have concerns regarding some of the new data:

(1) ImageNet results: Looking at table 9 in zico paper, we see that they achieve 77.1% imagenet accuracy with 600M FLOPS. However, in a comment to another reviewer, authors reveal that their MeCo found model requires nearly 884M FLOPS. This is a major increase in FLOPS with only 0.2% improvement in accuracy. Hence, this result is not convincing. Note that, there may be major differences between ZiCo and authors' training recipe and search space. If this is the case, I would encourage the authors to provide more details.

(2) The negative correlation thing and the channel sensitivity indeed seems to be a major weakness of this work. I appreciate authors' analysis that if c < wxh, MeCo would work better. However, this is violated for many image classification networks (e.g., on imagenet 224x224 input resolution with final layers getting 7x7 feature map size and much higher channel counts). The new results in Table 3 of the attached pdf also confirm that on some other use cases the MeCo does start losing correlations (although, on a lot of datasets, it performs very well). Hence, this can limit the applicability of this work. Can authors explore techniques to fix this channel sensitivity issue? Given this new information about c<wxh, it may actually be better if authors can mainly focus on vision tasks with much higher input-output resolutions, e.g., image super resolution and semantic segmentation. Specifically, for such vision applications, resolutions generally go up to 1080p or even 4K, so these applications may show the most benefits of authors' approach (and thus show that under applications with c<wxh, MeCo is always well behaved and does not lead to zero values, etc.). Of note, super resolution with smaller models (e.g., SESR (MLSys 2022), https://arxiv.org/abs/2103.09404) does not take long to train.

Anyways, given the improvement in clarity, new results, and new analysis, I am raising the rating to 4. I can raise the rating to 5 (or even 6 depending on how strong the results are) provided the authors (a) clearly identify the settings under which MeCo must be used (i.e., this dependence on c<wxh) right from the beginning of the paper (introduction, motivation, etc.), and (b) clearly demonstrate at least one vision application with high resolution input-output (e.g., super resolution...there have been NAS papers on this topic before: https://arxiv.org/pdf/2007.04356.pdf (ECCV 2020)). But without seeing major improvements on the above issues, I cannot increase the rating further at this time. I do think the authors are on to some interesting research. I would encourage them to continue this work and improve it further.

==== FURTHER UPDATE ====
Authors have fixed the main weakness with the introduction of their sensitivity correction method MeCo-Opt. All negative correlations are now positive (I do recommend to let other reviewers with lower rating know about this if they have similar concerns...but I leave that decision to the authors). A few Nas benches show near zero correlations but that is to be expected since no zero shot proxy is perfect. Moreover, the new zero shot proxy correction MeCo-Opt either maintains accuracy results or significantly improves them (e.g., for Imagenet-16-120 Nas bench). I hope the authors will use the remaining time until final notification to make sure their presentation is perfect in light of these new contributions: Specifically, motivate the need for metrics that work for higher resolution input-output vision tasks like segmentation, properly introduce the sensitivity correction in the main approach, update all the results with both MeCo and MeCo-Opt, conduct a few more real world experiments showing the significant improvements under MAC constraints that go beyond Nas benches (e.g., Imagenet and other Segmentation or Detection tasks) to complete the high-resolution input output picture and also provide conclusively better results over existing proxies under MAC constraints, and cover all the related work pointed out by all reviewers. The paper (particularly after rebuttal) has made strong contributions so please ensure that they are reflected in the presentation in the next version. I believe that with good presentation, this work should be of interest to the research community.

Given that my main concern is now resolved, I raise the rating to 7. Good luck!

**Questions:**

Please see above.

**Limitations:**

Yes, in appendix.

---

> ### Author Rebuttal · Authors · 2023-08-08
>
> Thanks a lot for your time and valuable comments and suggestions! We would like to respond to your reviews point by point.
>
> **Q1: Results on ImageNet.**
>
> Answer: We adopt MeCo-based NAS to ImageNet-1k dataset on MobileNet-v2 search space. The results are shown in Table 4 in the attached PDF. We trained the searched optimal network with 6 RTX 3090 for about 130 hours. The Top-1 accuracy is **77.3\%**, which is higher than all the baseline approaches.
>
> **Q2: Detailed analysis for the opposite results on TSS and SSS:**
>
> Answer:
> * Under what condition may the minimum eigenvalue become zero?
>
> In theory, for convolutional layers, $p_0$ is strictly greater than zero if $x_i \nparallel x_j, \forall I \neq j$ and $c<w \times h$, where $c$ is the \#channel, $w\times h$ is the input size. For ReLU activation, $p_0$ may be zero when the output matrix is singular.
>
> * In experiments, does $p_0=0$ (or $p_0$ is close to zero) affect our results?
>
> We would like to point out that this phenomenon will not affect the effectiveness of MeCo. Although MeCo may become (close to) zero when $c \ge w \times h$ due to the downsampling operation, all the architectures are compared under the same criteria. Specifically, in TSS, MeCo becomes zero only happens in the last set of cells where $c=w\times h=64$. However, all the topology has been evaluated in the previous cells in TSS. In SSS, MeCo is indeed sensitive to the #channels which leads to a strong negative correlation. However, the high correlation in Table 2 indicates that MeCo is still effective in discriminating and evaluating architectures in SSS. In fact, the most straightforward way to guarantee $c<w\times h$ is to change the input's dimension, making the size of the output tensor larger than #channels after all the downsampling. Thus we conduct the ablation study in Section 5 to show the performance of MeCo under different dimensions of inputs. The results in Figure 4 also support our theoretical analysis and shows that the best performance can be acquired when the input dimension is around $32 \times 32$ to $64 \times 64$. Finally, we would like to point out that the negative correlation on SSS also indicates a high correlation of #channels and accuracy on SSS search space.
>
> * Under what conditions would MeCo work optimally?
>
> The results of both the theory and experiments demonstrate that MeCo works best in evaluating the topology of the architectures (which again explains the good performance on TSS). To avoid MeCo from becoming zero and further improve its performance, we believe it is possible to limit #channels to satisfy the condition $c<w\times h$. This can be done by adding a sampling method of the output channels. We will leave this as future work.
> Finally, we found that in TSS, MeCo is zero in the last set of cells yet still achieves the highest correlation with the accuracy. This indicates that it is possible to evaluate the architectures *partially* without computing upon the whole skeleton. This is also intriguing and may further improve the performance of MeCo.
>
> **Q3: The ablation of the #Samples:**
>
> Answer: The ablation study in Section 5 is designed to demonstrate the relationship between MeCo and the dimension of the input samples, not the #Samples.
> We add ablation experiments about #Samples v.s. MeCo performance. We randomly chose different #Samples to evaluate the architectures in the entire TSS and SSS, respectively. The results are shown in Figure 1 in the attached PDF.
>
> It can be shown from the figure that MeCo is relatively stable when varying the #Samples. The performance of MeCo fluctuates around 0.01 throughout the experiments. These results support that MeCo only requires one data sample.
>
> **Q4: Variance of MeCo:**
>
> Answer: We have run MeCo 10 times from the same/different classes to see the variation, which is shown below:
>
> * Variation using samples from the same randomly chosen class
>
> || CIFAR-10| CIFAR-100| ImageNet16-120|
> |-|-|-|-|
> | NATSBench-TSS | 0.894 $\pm$ 0.003 | 0.886 $\pm$ 0.005 |0.845 $\pm$ 0.004|
> | NATSBench-SSS |-0.79 $\pm$ 0.02|-0.87 $\pm$ 0.01|-0.86 $\pm$ 0.02|
>
> * Variation using samples from different classes
>
> || CIFAR-10| CIFAR-100| ImageNet16-120|
> |-|-|-|-|
> | NATSBench-TSS | 0.893 $\pm$ 0.005 | 0.883 $\pm$ 0.005 |0.838 $\pm$ 0.007|
> | NATSBench-SSS |-0.79 $\pm$ 0.01|-0.86 $\pm$ 0.02|-0.86 $\pm$ 0.03|
>
> It can be seen that MeCo is robust when varying the input data and is irrelevant to the class label.
>
> **Q5: More results on other benchmarks:**
>
> Answer: We conduct experiments on Transbench-101, NASBench-301, NASBench-201, Transformer-like, and Mobilenet-like benchmarks. The results of the first three are shown in Table 1, 2, and 3 in the attached PDF.
>
> For Transformer-like benchmark AutoFormer, we load the trained supernets and regenerate the candidate subnets. We then re-evaluate the subnets to obtain the accuracy and compute MeCo. The correlation of MeCo and model accuracy on AutoFormer is **0.45**.
>
> For MobileNet-like benchmark OFA, we regenerate the architectures and directly compute MeCo upon the models. The correlation of MeCo and model accuracy on OFA is **0.864**, which is relatively high.
>
> It can be shown from the tables that MeCo achieves the highest correlation on Surface Normal+Macro, Class Objection+Micro, SCifar-100+Micro, NinaPro+Micro, NASBench-301+Cifar10, and NASBench-201+SVHN. MeCo also achieves competitive results on the majority of the remaining tasks. On some of the tasks, e.g., Autoencoding+Micro, NinaPro+201, MeCo is relatively low. This might be because of the characteristics of the task per se, and the channel-sensitive trait of MeCo. Designing proxies that can be further generalized to more tasks is a promising direction. We would like to note that the existing proxies do not achieve a high correlation on all task consistently.
>
> **Q6: Other papers in related work.**
>
> Answer: Thanks for your suggestions! We will add these papers to the related work and elaborate more.

---

> > ### Comment · Reviewer_4Zmk · 2023-08-19
> > **Thanks for the response. Raised the rating. Can increase further if the following weaknesses are fixed.**
> >
> > I have read the author response and other reviews. The new data and analyses provided by the authors have improved the paper compared to the initial submission. However, I still have concerns regarding some of the new data:
> >
> > (1) ImageNet results: Looking at table 9 in zico paper, we see that they achieve 77.1% imagenet accuracy with 600M FLOPS. However, in a comment to another reviewer, authors reveal that their MeCo found model requires nearly 884M FLOPS. This is a major increase in FLOPS with only 0.2% improvement in accuracy. Hence, this result is not convincing. Note that, there may be major differences between ZiCo and authors' training recipe and search space. If this is the case, I would encourage the authors to provide more details.
> >
> > (2) The negative correlation thing and the channel sensitivity indeed seems to be a major weakness of this work. I appreciate authors' analysis that if c < wxh, MeCo would work better. However, this is violated for many image classification networks (e.g., on imagenet 224x224 input resolution with final layers getting 7x7 feature map size and much higher channel counts). The new results in Table 3 of the attached pdf also confirm that on some other use cases the MeCo does start losing correlations (although, on a lot of datasets, it performs very well). Hence, this can limit the applicability of this work. Can authors explore techniques to fix this channel sensitivity issue? Given this new information about c<wxh, it may actually be better if authors can mainly focus on vision tasks with much higher input-output resolutions, e.g., image super resolution and semantic segmentation. Specifically, for such vision applications, resolutions generally go up to 1080p or even 4K, so these applications may show the most benefits of authors' approach (and thus show that under applications with c<wxh, MeCo is always well behaved and does not lead to zero values, etc.). Of note, super resolution with smaller models (e.g., SESR (MLSys 2022), https://arxiv.org/abs/2103.09404) does not take long to train.
> >
> > Anyways, given the improvement in clarity, new results, and new analysis, I am raising the rating to 4. I can raise the rating to 5 (or even 6 depending on how strong the results are) provided the authors (a) clearly identify the settings under which MeCo must be used (i.e., this dependence on c<wxh) right from the beginning of the paper (introduction, motivation, etc.), and (b) clearly demonstrate at least one vision application with high resolution input-output (e.g., super resolution...there have been NAS papers on this topic before: https://arxiv.org/pdf/2007.04356.pdf (ECCV 2020)). But without seeing major improvements on the above issues, I cannot increase the rating further at this time. I do think the authors are on to some interesting research. I would encourage them to continue this work and improve it further.

---

> > > ### Author Response · Authors · 2023-08-20
> > > **Response to the Follow-up Questions**
> > >
> > > Dear Reviewer 4Zmk,
> > >
> > > Thanks for your valuable feedback. We would like to respond to your concerns as follow.
> > >
> > > **Q1: ImageNet training details.**
> > >
> > > We follow the training settings in Zico since it is the best baseline currently. The difference is that we did not limit the FLOPs budget because the generation strategy we used is an unconstrained NAS. Other than that, we did not add additional augmentation methods.
> > >
> > > **Q2: Channel Sensitivity Issue.**
> > >
> > > We would like to address this issue from the following key points.
> > >
> > > * How to solve the channel-sensitivity of MeCo?
> > >
> > > We have been thinking about addressing the channel-sensitivity issue. We would like to introduce a weighted-sampling method as an optimization of the original MeCo.
> > >
> > > Specifically, for the feature map of layer $l$, the original MeCo flattens each channel of the feature map and computes $p_0$ as the evaluation result. This operation makes MeCo be sensitive to \#channels, e.g., the larger \#channels, the smaller $p_0$, even $p_0=0$. To address this issue, alternatively, we randomly **sample a fixed number of channels** and flatten them to compute $p_0'$. We then compute the final result by **multiplying $p_0'$ with a channel weight**, such that for layer $l$,
> > >
> > >
> > > $$
> > > \mathsf{MeCo}_{\mathsf{opt}}^{l} := \frac{c}{n} \cdot p_0'
> > > $$
> > >
> > > where $c$ is #channels, $n$ is the fixed sampling numbers.
> > >
> > > The high-level idea of this optimization is to limit the dimension of the Pearson correlation matrix by constraining #channels. Instead of computing upon a large matrix, we calculate the minimum eigenvalue upon a fixed-sized matrix and then enlarge it with corresponding constants. Note that the bonus of this optimization is that the time-cost can be controlled because the matrix dimension is significantly reduced compared to the original one.
> > >
> > > * What is the performance of the MeCo with optimization?
> > >
> > > We utilize $\mathsf{MeCo}_\mathsf{opt}$ on NATS-Bench-TSS and NATS-Bench-SSS to show the Spearman correlation $\rho$. We set $n=8$. The results are shown below.
> > >
> > > || Cifar-10| Cifar-100| ImageNet16-120|
> > > |-|-|-|-|
> > > | TSS | 0.90 $\pm$ 0.002 | 0.89 $\pm$ 0.003 | 0.85 $\pm$ 0.003 |
> > > | SSS | 0.89 $\pm$ 0.002 | 0.83 $\pm$ 0.004 | 0.89 $\pm$0.003|
> > >
> > > It can be shown from the table that the weighted-sampling method is effective in fixing the channel-sensitive issue. $\mathsf{MeCo}_\mathsf{opt}$ shows a strong **positive** correlation with SSS accuracy, while the TSS results are also promoted. The correlations on both TSS and SSS are the highest among all the baseline proxies.
> > >
> > > * What is the performance of MeCo on high-resolution input-output tasks?
> > >
> > > We thank the reviewer's insight. We would like to show the performance of MeCo before and after optimization on **Segmentation** task in Transbench101 Micro and Macro.
> > >
> > > || MeCo-$\rho$ | MeCo-opt-$\rho$ | MeCo-acc | MeCo-opt-acc |
> > > |-|-|-|-|-|
> > > | Micro | 0.62| 0.63| 94.88| 94.97|
> > > | Macro | 0.61| 0.77| 95.30| 95.13|
> > >
> > > The correlation $\rho$ has been increased after adopting the optimization, and the searched optimal model achieves around 95\% accuracy. All the optimal models are trained on Taskonomy dataset under a learning rate of 0.1 and softmax+CE loss function. The output dimension is $256 \times 256$.
> > >
> > > * How to revise our paper?
> > >
> > > Based on the above results and suggestions from the reviewer, we present the following plan to revise our paper.
> > >
> > > In ``Introduction``, we will add a "Remark" paragraph to clearly demonstrate that our original MeCo has limited ability to evaluate networks when the number of channels is larger than the feature map dimension. We suggest the original MeCo be adopted for high-resolution input-output scenarios such as the segmentation task. We will then point out that we further propose an optimization approach to alleviate this issue, making MeCo a general proxy.
> > >
> > > We will add a ``Discussion and Optimization`` section after the ``Experiments`` part to analyze the channel-sensitive issue of the original MeCo. We clarify that the original MeCo is suitable for high-resolution input-output tasks like Segmentation. We present the results of segmentation here as elaboration. We will then present the weighted-sampling optimization. The updated correlation on TSS and SSS will also be added to illustrate the effectiveness of the optimization.
> > >
> > > We did not focus MeCo entirely on vision tasks since we proposed a possible optimization to alleviate the channel-sensitive issue. We appreciate the reviewer's suggestions and sincerely hope our solution for revising the paper is acceptable.

---

> > > > ### Comment · Reviewer_4Zmk · 2023-08-21
> > > > **A few other follow up questions**
> > > >
> > > > Thank you for new experiments. A few other questions:
> > > > 1. How to select hyperparameter n in this new MeCo-Opt method? If theoretical justification is not possible at this point, I'd appreciate any empirical guidelines that may help practitioners.
> > > > 2. Why does Macro accuracy reduce on the segmentation task from 95.30% to 95.13% even though correlation increases from 0.61 to 0.77 for the macro case? Is there any stochastic behavior in the search process that may be affecting results? (I do not expect stochastic behavior since it is a nas bench result).
> > > > 3. Can authors show all nas bench results with the new MeCo-Opt method to show that this fixes (preferably with a single fixed hyperparameter value for n) all negative correlation issues, without reducing the performance on Nas benches where original MeCo was already giving good results? Ideally, I'd also like to see some accuracy numbers in real world experiments/search spaces on imagenet and other vision tasks to see if this improves real world accuracy numbers compared to the ZiCo, etc. I understand that it might be too late to produce those numbers right now. So, at the very least, it would be useful to see accuracy numbers from nas benches with the new MeCo-Opt score. For example, can authors show: (A) all negative and zero correlations become strongly positive, and all existing positive corrections do not suffer. (B) accuracy results from search (on Nas benches or real datasets, whichever is possible) significantly improve with the new MeCo-Opt method (particularly for cases where the original method was giving negative or zero correlations. And how do these new accuracy numbers compared to other sota zero shot metrics?
> > > >
> > > > If results from the new score are significant, I'd suggest authors to add another general comment to all reviewers...in case they want to take that into account.

---

> > > > > ### Author Response · Authors · 2023-08-21
> > > > > **Respond to Follow-up Questions (1/2)**
> > > > >
> > > > > Dear Reviewer 4Zmk,
> > > > >
> > > > > Thanks for your feedback. We would like to answer your questions as follows.
> > > > >
> > > > > Q1: Hyperparameter selection
> > > > >
> > > > > MeCo-opt aims to solve the channel-sensitive issue, and $p_0$ might be zero due to $c \ge w \times h$. Thus when choosing the fixed number of channels $n$, we require $n$ to be smaller than the minimum size of the feature map. Moreover, $n$ should be smaller than the miminum \#channels of the architectures. Under these conditions, we set $n$ as large as possible to retain more information.
> > > > >
> > > > > We conduct a small ablation study on $n$ v.s. MeCo_opt on TSS and SSS. The results are shown below.
> > > > >
> > > > > * TSS
> > > > >
> > > > > || Cifar-10| Cifar-100| ImageNet16-120|
> > > > > |-|-|-|-|
> > > > > | $n=4$ | 0.87 $\pm$ 0.003| 0.88 $\pm$ 0.002| 0.84 $\pm$ 0.005|
> > > > > | $n=6$ | 0.88 $\pm$ 0.004 | 0.88 $\pm$ 0.001| 0.84 $\pm$ 0.004|
> > > > > | $n=8$ | 0.90 $\pm$ 0.002 | 0.89 $\pm$ 0.003 | 0.85 $\pm$ 0.003 |
> > > > >
> > > > > * SSS
> > > > >
> > > > > || Cifar-10| Cifar-100| ImageNet16-120|
> > > > > |-|-|-|-|
> > > > > | $n=4$ | 0.88 $\pm$ 0.005| 0.82 $\pm$ 0.003 | 0.84 $\pm$ 0.003|
> > > > > | $n=6$ | 0.88 $\pm$ 0.007 | 0.83 $\pm$ 0.002 | 0.85 $\pm$ 0.006 |
> > > > > | $n=8$ | 0.89 $\pm$ 0.002 | 0.83 $\pm$ 0.004 | 0.89 $\pm$0.003|
> > > > >
> > > > >
> > > > > Q2: Accuracy and correlation on Transbench-101-Macro.
> > > > >
> > > > > The correlation with the accuracy is a statistical result for a specific benchmark. A higher correlation does not always lead to higher accuracy for the **best** architecture. There is no stochastic behavior involved in the results.

---

> > > > > ### Author Response · Authors · 2023-08-21
> > > > > **Respond to Follow-up Questions (2/2)**
> > > > >
> > > > > Q3: MeCo-Opt on other benchmarks.
> > > > >
> > > > > 1. Correlation results
> > > > >
> > > > > We present the correlation results of MeCo-opt on NasBench-301, Transbench101-Micro, and Transbench101-Macro. **All the experiments are conducted under $n=8$.**
> > > > >
> > > > > * NASBench301
> > > > >
> > > > > |Method|Cifar-10|Spherical-Cifar-100|NinaPro|SVHN|
> > > > > |-|-|-|-|-|
> > > > > |grasp|0.34|**0.13**|0.04|0.18|
> > > > > |fisher|-0.28|0.00|-0.11|0.05|
> > > > > |grad_norm|-0.04|-0.00|**-0.20**|0.42|
> > > > > |snip|-0.05|-0.01|-0.10|0.38|
> > > > > |synflow|0.18|0.05|-0.07|0.50|
> > > > > |l2_norm|0.45|0.12|-0.07|**0.70**|
> > > > > |#params|0.46|0.07|-0.07|**0.70**|
> > > > > |zen|0.43|0.07|-0.09|0.68|
> > > > > |jacov|-0.04|0.08|0.13|-0.36|
> > > > > |nwot|0.47|0.05|0.02|0.64|
> > > > > |zico|0.66|-|-|-|
> > > > > |MeCo (Ours)|0.70|-0.05|-0.11|0.68|
> > > > > |**MeCo-opt**|**0.71**|**0.03**|**0.12**|**0.68**|
> > > > >
> > > > > * Transbench101-Macro
> > > > >
> > > > > |Method|Autoencoder|Class Objection|Scene Classification|Jigsaw|Surface Normal|Segmantation|Room Layout|
> > > > > |-|-|-|-|-|-|-|-|
> > > > > |grasp|-0.02|-0.64|-0.43|-0.26|-0.05|-0.02|-0.26|
> > > > > |fisher|-0.19|-0.30|-0.13|-0.26|0.15|0.03|-0.26|
> > > > > |grad_norm|0.31|-0.56|-0.33|-0.27|0.35|0.21|-0.27|
> > > > > |snip|0.20|-0.38|-0.14|-0.19|0.45|0.27|-0.19|
> > > > > |synflow|0.00|0.12|0.27|0.34|0.00|0.00|0.34|
> > > > > |l2_norm|-0.20|0.08|0.28|0.15|0.30|0.18|0.15|
> > > > > |#params|-0.18|0.16|0.32|0.15|0.30|0.06|0.15|
> > > > > |zen|-0.01|0.10|0.27|0.24|0.38|0.27|0.24|
> > > > > |jacov|0.45|0.07|0.19|0.19|0.50|0.57|0.19|
> > > > > |nwot|0.67|**0.83**|**0.89**|0.48|0.78|**0.80**|**0.76**|
> > > > > |MeCo (Ours)|0.51|0.59|0.81|0.17|0.80|0.62|0.23|
> > > > > |**MeCo-opt**|**0.74**|**0.73**|**0.76**|**0.48**|**0.76**|**0.63**|**0.33**|
> > > > >
> > > > > * Transbench101-Micro
> > > > >
> > > > > |Method|Autoencoding|Class Objection|Scene Classification|Jigsaw|Surface Normal|Segman- tation|Room Layout|Spherical-Cifar-100|NinaPro|SVHN|
> > > > > |-|-|-|-|-|-|-|-|-|-|-|
> > > > > |grasp|-0.12|-0.22|-0.43|-0.12|0.01|0.00|-0.29|-0.03|-0.20|-0.24|
> > > > > |fisher|**-0.58**|0.44|-0.13|0.30|0.16|0.12|0.30|0.72|0.42|0.81|
> > > > > |grad_norm|-0.32|0.39|-0.33|0.36|0.36|0.60|0.25|0.72|0.40|0.78|
> > > > > |snip|-0.27|0.45|-0.14|0.41|0.49|0.68|0.32|0.76|0.42|0.83|
> > > > > |synflow|0.00|0.48|0.27|0.47|0.00|0.00|0.30|0.79|0.45|**0.92**|
> > > > > |l2_norm|0.04|0.32|0.28|0.35|0.50|0.48|0.18|0.53|0.36|0.52|
> > > > > |#params|-0.01|0.45|0.32|0.44|0.62|0.68|0.30|0.79|0.36|0.76|
> > > > > |zen|0.14|0.54|0.27|0.51|0.71|0.67|0.38|0.67|0.42|0.74|
> > > > > |jacov|0.18|0.51|0.19|**0.56**|**0.75**|**0.80**|**0.40**|0.71|0.40|0.77|
> > > > > |nwot|0.03|0.39|**0.89**|0.42|0.57|0.53|0.25|0.64|0.38|0.63|
> > > > > |zico|0.35|-|0.71|0.52|0.68|-|-|-|-|-|
> > > > > |MeCo (Ours)|0.03|0.58|0.62|0.45|0.65|0.62|-0.25|0.85|0.47|0.88|
> > > > > |**MeCo-opt**|**0.03**|**0.59**|**0.64**|**0.47**|**0.67**|**0.77**|**0.26**|**0.85**|**0.47**|**0.88**|
> > > > >
> > > > > The results show that MeCo-opt effectively solves the negative correlation issue on NASBench301 and Room+Micro. The majority of the positive correlations are promoted up to 0.23, and the remaining results are not impacted.
> > > > >
> > > > > 2. Accuracy of the searched optimal model
> > > > >
> > > > > We provide the accuracy of the searched optimal model on TSS and SSS for ten runs. We provide the accuracy of the *baseline methods that are higher than ours* for comparison. The performance of the remaining baselines can be found in Tab.4 in the supplementary materials.
> > > > >
> > > > > * TSS
> > > > >
> > > > > || Cifar-10| Cifar-100| ImageNet16-120|
> > > > > |-|-|-|-|
> > > > > |ZiCo| 93.56 $\pm$ 0.23 |70.64 $\pm$ 0.28 | 42.74 $\pm$ 1.78|
> > > > > |synflow| 93.27 $\pm$ 0.74 |**71.12 $\pm$ 1.59** | 42.65 $\pm$ 3.59|
> > > > > |MeCo|**93.64 $\pm$ 0.31**| 70.86 $\pm$ 0.96| 42.59 $\pm$ 1.77|
> > > > > |MeCo-opt| 93.63 $\pm$ 0.26 | 70.79 $\pm$ 0.35 | **42.84 $\pm$ 1.60**|
> > > > >
> > > > > * SSS
> > > > >
> > > > > || Cifar-10| Cifar-100| ImageNet16-120|
> > > > > |-|-|-|-|
> > > > > |ZiCo| 93.06 $\pm$ 0.23 | 69.65 $\pm$ 0.54 | 45.40 $\pm$ 0.72|
> > > > > |MeCo| 93.12 $\pm$ 0.24 | **69.89 $\pm$ 0.42** | 44.34 $\pm$ 1.78|
> > > > > |MeCo-opt| **93.37 $\pm$ 0.19** | 69.75 $\pm$ 0.35 | **46.62 $\pm$ 0.70**|
> > > > >
> > > > > We would like to point out that although MeCo is sensitive to \#channels and shows a negative correlation on SSS, it is still effective in evaluating the networks. Moreover, the optimization method MeCo-opt is effective in promoting performance.
> > > > >
> > > > > We will add the above ablation study and new experimental results to the corresponding tables in our paper and elaborate more in the ``Discussion and Optimizations`` section.
> > > > >
> > > > > Finally, we would like to re-emphasize the main contributions of this paper since the discussion period is ending soon. In this paper, we propose a new proxy that **only requires one data sample in one forward pass**. We provide theoretical analysis regarding both the convergence rate and generalization capacity. MeCo obtains the highest correlation on TSS, SSS, 301, and some tasks in Transbench101. We acknowledge that MeCo is sensitive to the #channels, yet **the negative correlation does not indicate the ineffectiveness of MeCo**. To further alleviate this issue, we propose an optimization method MeCo-opt. The performance of MeCo-opt is also provided, which is effective.
> > > > >
> > > > > We thank the reviewer's patients and constructive comments in such a short period. We sincerely hope the results can address the reviewer's concerns and could lead to a positive rating of our paper.

---

> > > > > > ### Comment · Reviewer_4Zmk · 2023-08-21
> > > > > > **Final update. Rating raised as promised.**
> > > > > >
> > > > > > Authors have fixed the main weakness with the introduction of their sensitivity correction method MeCo-Opt. All negative correlations are now positive (I do recommend to let other reviewers with lower rating know about this if they have similar concerns...but I leave that decision to the authors). A few Nas benches show near zero correlations but that is to be expected since no zero shot proxy is perfect. Moreover, the new zero shot proxy correction MeCo-Opt either maintains accuracy results or significantly improves them (e.g., for Imagenet-16-120 Nas bench). I hope the authors will use the remaining time until final notification to make sure their presentation is perfect in light of these new contributions: Specifically, motivate the need for metrics that work for higher resolution input-output vision tasks like segmentation, properly introduce the sensitivity correction in the main approach, update all the results with both MeCo and MeCo-Opt, conduct a few more real world experiments showing the significant improvements under MAC constraints that go beyond Nas benches (e.g., Imagenet and other Segmentation or Detection tasks) to complete the high-resolution input output picture and also provide conclusively better results over existing proxies under MAC constraints, and cover all the related work pointed out by all reviewers. The paper (particularly after rebuttal) has made strong contributions so please ensure that they are reflected in the presentation in the next version. I believe that with good presentation, this work should be of interest to the research community.
> > > > > >
> > > > > > Given that my main concern is now resolved, I raise the rating to 7. Good luck!

---

> > > > > > > ### Author Response · Authors · 2023-08-21
> > > > > > > **Thanks for increasing the score!**
> > > > > > >
> > > > > > > Dear Reviewer 4Zmk,
> > > > > > >
> > > > > > > We appreciate all the valuable and constructive comments which significantly improve the quality of our paper. We will further add a general response and carefully revise our paper according to the reviews in the rebuttal and discussion phases.
> > > > > > >
> > > > > > > Thanks again and best wishes,
> > > > > > >
> > > > > > > Authors 7200

---

### Official Review · Reviewer_5B3N · 2023-07-06

**Soundness:** 3 good
**Presentation:** 3 good
**Contribution:** 3 good
**Rating:** 6
**Confidence:** 4

**Summary:**

This paper proposes a new proxy ie MeCo for zero-cost neural architecture search. The proxy design aims at overcoming two important disadvantages of zero-cost proxies proposed earlier. The first one is that these proxies invoke at least one back prop and hence are still expensive. The second disadvantage is that the proxies are very much dependent on the data and labels considered.  The designed proxy is evaluated on NATS-Bench-TSS  and NATS Bench-SSS benchmarks and achieves a very high correlation with the ground truth in comparison to other SOTA proxies. Further integration of MeCO can be easily integrated with existing generation method for complete NAS and shows performance gains in comparison to other zero cost proxies. The code of the paper is open-sourced making the approach reproducible.


**Strengths:**

The paper is very well motivated, well-written and clear. The theoretical motivation behind the MeCO is very clearly established. The idea behind the designed proxy is fairly novel to NAS and its effectiveness is established across different benchmarks and also for complete NAS. The authors adequately discuss their hyperparameter settings and release their code for reproducibility. The evaluation is fairly thorough on the NATS-Bench and the method is well-ablated.

**Weaknesses:**

Main weakness of the paper in my opinion is the search spaces the proxy is studied on.  [NAS-Bench Suite Zero](https://arxiv.org/pdf/2210.03230.pdf) provides easy access and evaluation on about 13 proxies and 28 tasks. Since the performance of a given proxy can vary widely depending upon the search space, task and datasets I am not convinced of the effectiveness of the proxy based on only the results on NATS-Bench and DARTS spaces. Furthermore how would one apply proxies to transformer spaces (eg [AutoFormer](https://openaccess.thecvf.com/content/ICCV2021/papers/Chen_AutoFormer_Searching_Transformers_for_Visual_Recognition_ICCV_2021_paper.pdf) , [HAT](https://arxiv.org/abs/2005.14187)) and MobileNets(eg: [OFA](https://arxiv.org/pdf/1908.09791.pdf),) which provide queryable validation accuracy based on a surrogate. Can this proxy be applied in these spaces? Since modern one-shot NAS methods are applied on transformer and mobilenet spaces too, it is important to design a proxy that does indeed generalize well. I encourage the authors to evaluate their proxy on these search spaces too.

**Questions:**

In addition to comments in the weakness section I have the following questions:
1. Could you evaluate the proxy on all tasks from NAS-Bench suite zero and transformer spaces?
2. Why is the correlation coefficient negative on NATS-Bench-SSS? Could you please elaborate the explanation from line 228-246
3. Could you compare the cost in computing MeCo v/s other zero-cost proxies?


**Limitations:**

I encourage to discuss the limitations of the designed proxy adequately, for example address the following points:
1. Are there any important assumptions made in designing the proxy, which may not hold in practice?
2. Evaluation of how good a zero cost proxy is, is often tied to availability of benchmarks it can be evaluated on. Hence though the proxies 3. are indeed “zero-cost”, the evaluation of the proxy quality is strongly tied to a benchmark which in the first place is very expensive to create. It would be great if the authors could comment on this in the limitations too.

---

> ### Author Rebuttal · Authors · 2023-08-08
>
> Thanks a lot for your time and valuable comments and suggestions! We would like to respond to your reviews point by point.
>
> **Q1: Could you evaluate the proxy on all tasks from NAS-Bench suite zero and transformer spaces?**
>
> Answer: Thanks for your suggestions! We conduct the experiments on all 28 tasks mentioned in NAS-Bench-Zero-Suite. For character limits, we present the results in the attached PDF, in which Table 1 is 10 tasks on Transbench-101-Micro, Table 2 is 7 tasks on Transbench-101-Macro, Table 3 is 4 tasks on NASBench-301 and 3 tasks on NASBench-201. The remaining 3 tasks on NASBench-201 and 1 task on NASBench-101 are presented in our paper.
>
> It can be shown from the tables that MeCo achieves the highest correlation on Surface Normal+Macro, Class Objection+Micro, SCifar-100+Micro, NinaPro+Micro, NASBench-301+CIFAR10, and NASBench-201+SVHN. MeCo also achieves competitive results on the majority of the remaining tasks. On some of the tasks, e.g., Autoencoding+Micro, NinaPro+201, MeCo is relatively low. This might be because of the characteristics of the task per se, and the channel-sensitive trait of MeCo. Designing proxies that can be further generalized to more tasks is a promising direction. We would like to note that the existing proxies do not achieve a high correlation on all tasks consistently.
>
> **Q2: Why is the correlation coefficient negative on NATS-Bench-SSS? Could you please elaborate the explanation from line 228-246?**
>
> Answer: In our paper 3.2.3, we summarize the main theoretical conclusions in which we point out that a multi-channel convolutional layer can be viewed as a multi-sample over-parameterized NN layer. This leads to the fact that MeCo is sensitive to the #channels (which will be treated as samples in MeCo). The more channels contained, the larger the Pearson matrix, which will lead to SMALLER $p_0$. The simulated results in Figure 2 also validate this issue. On the other hand, larger #channels will normally lead to higher accuracy of the architectures (yet lower convergence rate). Thus, the variation of #channels in SSS (chosen from 8 to 64) leads to a negative correlation on SSS. In TSS, however, all the networks share the same #channels at each stage, thus MeCo does not show a negative correlation.
>
> We would like to further point out that this phenomenon does not contradict our theoretical basis. In practice, various factors (e.g., #channels, #samples, parameters) may affect the network's accuracy. Theorem 2 and Theorem 3 (provided in rebuttal) shed light on the relationship between $p_0$ and the convergence rate/generalization capacity of the network, which has been proven to be effective in multiple benchmarks.
>
> **Q3: Could you compare the cost in computing MeCo v/s other zero-cost proxies?**
>
> We present the time cost in computing MeCo and other baseline proxies for all architectures in NASBench-201 and 10k networks in NASBench-301.
>
> * NASBench-201 (GPU Hours):
>
> ||MeCo (Ours)|fisher|grad_norm|grasp|jacov|nwot|params|snip|synflow|zen|ZiCo|
> |-|-|-|-|-|-|-|-|-|-|-|-|
> |CIFAR10|**2.82**|4.65|4.45|13.59|4.46|3.53|1.56|4.48|2.80|3.51|3.17|
> |CIFAR100|**2.82**|4.68|4.43|13.58|4.44|3.50|1.59|4.47|2.18|3.50|3.73|
> |ImageNet16-120|**2.83**|4.51|4.34|6.62|4.38|3.94|2.62|4.37|3.82|3.99|3.71|
>
> * NASBench-301 (GPU Hours):
>
> | |MeCo (Ours)|fisher|grad_norm|grasp|jacov|nwot|params|snip|synflow|zen|
> |--|--|--|--|--|--|--|--|--|--|--|
> |CIFAR10|**2.52**|7.33|6.77|32.74|7.05|3.83|1.54|6.67|3.18|4.44|
>
> MeCo is efficient because it only requires one forward pass. Although it is slower than #Params, MeCo achieves a higher correlation than #Params on multiple benchmarks.
>
> **Q4 (Other concerns in ``Weakness`` part): How would one apply proxies to transformer spaces (eg AutoFormer , HAT) and MobileNets(eg: OFA,)? Can this proxy be applied in these spaces?**
>
> We conduct the experiments on the transformer search space (AutoFormer) and MobileNets-like search space (OFA).
>
> For AutoFormer, we load the trained supernets and regenerate the candidate subnets. We then re-evaluate the subnets to obtain the accuracy and compute MeCo. The correlation of MeCo and model accuracy on AutoFormer is **0.45**.
>
> For MobileNet OFA, we regenerate the architectures and directly compute MeCo upon the models. The correlation of MeCo and model accuracy on OFA is **0.864**, which is relatively high.
>
> P.S., we use the AutoFormer[1] instead of the paper provided in the review link, which we carefully checked and thought it might be another paper with the same name :)
>
> **Q5 (Other concerns in ``Limitation`` part): Are there any important assumptions made in designing the proxy, which may not hold in practice?**
>
> Answer: In our theoretical analysis, we assume there are no parallel inputs and $ c<w \times h $, where $c$ is the number of channels, $w\times h$ is the size of the input. Under this assumption, $p_0$ is strictly greater than zero. This might not hold in practice when there are parallel inputs or large #channels. We will elaborate on this issue in the ``Limitation`` section.
>
> **Q6 (Other concerns in ``Limitation`` part): Valuation of how good a zero-cost proxy is, is often tied to availability of benchmarks it can be evaluated on. Hence though the proxies 3. are indeed “zero cost”, the evaluation of the proxy quality is strongly tied to a benchmark which in the first place is very expensive to create. It would be great if the authors could comment on this in the limitations too.**
>
> Answer: Thanks for your valuable suggestion. We will further comment on this in the ``Limitation`` Section.
>
> [1] Chen M, Peng H, Fu J, et al. Autoformer: Searching transformers for visual recognition[C]//Proceedings of the IEEE/CVF international conference on computer vision. 2021: 12270-12280.

---

> > ### Comment · Reviewer_5B3N · 2023-08-15
> >
> > I thank the authors for their detailed response and further evaluations. The response addresses most of my concerns. I have updated my score

---

> > > ### Author Response · Authors · 2023-08-15
> > > **Thanks for Increasing the Score!**
> > >
> > > Dear Reviewer 5B3N,
> > >
> > > We deeply appreciate your time, valuable suggestions, and acknowledgment of our previous work!
> > >
> > > Thanks a lot and best wishes!
> > >
> > > Authors of paper 7200

---

### Official Review · Reviewer_BGPn · 2023-07-06

**Soundness:** 4 excellent
**Presentation:** 3 good
**Contribution:** 3 good
**Rating:** 7
**Confidence:** 4

**Summary:**

This paper introduce a new zero-cost proxy for NAS, called MeCo, which is fully independent of data and labels and requires only one data sample for a single forward pass. The paper provides a theoretical analysis of the relationship between the Pearson correlation matrix of the feature maps and the training convergence rate of the network. Meco shows high ranking correlation in NATS-Bench.

**Strengths:**

The authors theoretically reveal that the minimum eigenvalue of the correlation on the feature map is correlated with the training convergence rate. Based on this analysis, they propose Meco as a novel training-free proxy. Experiments conducted on NATS-Bench demonstrate the superiority of Meco for three small datasets.

**Weaknesses:**

1. As acknowledged by the authors, Meco is sensitive to variations in the number of channels, implying that Meco exhibits a similar preference as #Param when selecting an architecture from the search space. However, the paper lacks the ranking correlation of Param in Tab1 and Tab2, making it difficult to assess the correlation between Meco and Param. Besides the marginal improvements in ranking correlation, it would be beneficial to understand the advantages of Meco over Param.

2. It is recommended to address the variation of correlation in Tab1 and Tab2. Many zero-cost proxies can experience fluctuations in ranking correlation when different batches with the same batch size are used. To showcase the robustness of Meco, providing the variation would be valuable.

3. I have a question regarding the NAS-Benchmarks. The abstract states that NAS-Bench-101 is used in this paper; however, there are no experiments conducted on NAS-Bench-101 in the main body, only in the supplementary material. It is inappropriate to claim that experiments were performed on NB101 without presenting the results in the main body. Additionally, NAS-Bench-201 is a widely-used benchmark in the NAS community. I noticed that there are some files related to NB201 in the provided code. Why was NAS-Bench-201 not utilized?

4. I noticed an inconsistency in the ranking correlation of NTK between Table3 (0.78) and Table1 (0.76). Please provide an explanation for this discrepancy.

5. It should be noted that in recent years, there have been advancements in zero-shot proxies, including NASWOT, KNAS, NASI, and GradSign. However, this paper only focuses on older methods and does not consider these newer counterparts.

6. I am curious about the computational efficiency of Meco when using `torch.corrcoef(fea)`. In practical neural networks, especially in later layers, the channel number can be quite large. Therefore, I am questioning whether Meco can be extended to larger networks with a large number of channels while maintaining computational efficiency.

**Questions:**

What are the advantages of Meco over #Param, besides the marginal improvements in ranking correlation? Is there a high correlation between Meco and #Param?

How does the correlation variation of Meco address the issue of fluctuation in ranking correlation seen in other zero-cost proxies? Can the robustness of Meco be demonstrated through providing the correlation variation?

Why does the paper claim to use NAS-Bench-101 but not provide the experimental results in the main body? Why was NAS-Bench-201, a widely-used benchmark in the NAS community, not utilized?

Can Meco maintain computational efficiency when applied to larger networks with a large number of channels, considering the practical implications for neural networks?


**Limitations:**

I have concerns regarding the experimental aspects of the paper. During the rebuttal period, I hope the authors can address these concerns.

---

> ### Author Rebuttal · Authors · 2023-08-08
>
> Thanks a lot for your time and valuable comments and suggestions! We would like to respond to your reviews point by point.
>
> **Q1: What are the advantages of Meco over #Param, besides the marginal improvements in ranking correlation? Is there a high correlation between Meco and #Param?**
>
> Answer: We demonstrate the performance of #Param on NATSBench-TSS and NATSBench-SSS as follows:
>
> * NATSBench-TSS:
>
> || CIFAR-10 | CIFAR-100 | ImageNet16-120 |
> |-|-|-|-|
> | #Param| 0.72| 0.73| 0.69|
> | MeCo (Ours) | 0.89| 0.88| 0.85|
>
> * NATSBench-SSS:
>
> || CIFAR-10 | CIFAR-100 | ImageNet16-120 |
> |-|-|-|-|
> | #Param| 0.72| 0.73| 0.84|
> | MeCo (Ours) | -0.80| -0.89| -0.88|
>
> MeCo is proved to be better than #Param on NATSBench-TSS and NATSBench-SSS benchmarks. We present the correlation between MeCo and #Param as follows:
>
> || CIFAR-10 | CIFAR-100 | ImageNet16-120 |
> |-|-|-|-|
> | NATSBench-TSS| 0.83| 0.83| 0.85|
> | NATSBench-SSS| -0.72| -0.69| -0.68|
>
> MeCo shows a relatively strong correlation with #Param. Other than the higher correlation on various benchmarks, MeCo achieves much higher Top-1 accuracy than #Params on large-scaled datasets such as ImageNet-1k (77.3\% v.s. 63.5\% on MobileNet-v2). Finally, due to the theoretical basis of MeCo, we believe it can be further optimized and extended to a zero-shot NAS with better performance.
>
>
> **Q2: How does the correlation variation of Meco address the issue of fluctuation in ranking correlation seen in other zero-cost proxies? Can the robustness of Meco be demonstrated through providing the correlation variation?**
>
> Answer: We have executed MeCo 10 times from the same/different classes to see the variation, which is shown below:
>
> * Variation using samples from the same randomly chosen class
>
> || CIFAR-10| CIFAR-100| ImageNet16-120|
> |-|-|-|-|
> | NATSBench-TSS | 0.894 $\pm$ 0.003 | 0.886 $\pm$ 0.005 |0.845 $\pm$ 0.004|
> | NATSBench-SSS |-0.79 $\pm$ 0.02|-0.87 $\pm$ 0.01|-0.86 $\pm$ 0.02|
>
> * Variation using samples from different classes
>
> || CIFAR-10| CIFAR-100| ImageNet16-120|
> |-|-|-|-|
> | NATSBench-TSS | 0.893 $\pm$ 0.005 | 0.883 $\pm$ 0.005 |0.838 $\pm$ 0.007|
> | NATSBench-SSS |-0.79 $\pm$ 0.01|-0.86 $\pm$ 0.02|-0.86 $\pm$ 0.03|
>
> It can be seen that MeCo is robust when varying the input data and is irrelevant to the class label.
>
> **Q3: Why does the paper claim to use NAS-Bench-101 but not provide the experimental results in the main body? Why was NAS-Bench-201, a widely-used benchmark in the NAS community, not utilized?**
>
> Answer: We did not provide NAS-Bench-101 in the main body due to the page limits, which is inappropriate to put in the appendix while mentioned in the abstract (thanks for pointing out). We consider moving the results to the main body if the page limit permits, or does not point out NAS-Bench-101 in the abstract part.
>
> We did conduct experiments on NAS-Bench-201, which is represented as “NATSBench-TSS” in our paper. We also evaluated MeCo on the extended version of NAS-Bench-201 and denoted as "SSS". We will clarify the name of the benchmarks to avoid possible confusion.
>
> **Q4: Can Meco maintain computational efficiency when applied to larger networks with a large number of channels, considering the practical implications for neural networks?**
>
> Answer: MeCo can be adapted to larger networks with more channels because it does not rely on the gradients and only needs one input data. For example, MeCo requires 2.52 GPU hours to evaluate 10k networks in NAS-Bench-301. In practice, the main computation bottleneck is the eigenvalue calculation of the feature maps. For now, we execute MeCo without any thread-level parallelization or optimizations. Moreover, it is possible to constrain the number of channels to make MeCo more efficient. In all, the efficiency of MeCo when applied to larger networks can be further promoted using parallelization and optimized approaches in real applications.
>
> **Q5 (Other concerns in the *Weakness* part): inconsistency in the ranking correlation of NTK between Tab 3 (0.78) and Tab 1 (0.76):**
>
> Answer: The results in Table 1 are obtained using one data sample from the corresponding dataset (i.e., CIFAR-10/CIFAR-100/ImageNet16-120). While the results in Table 3 are obtained by random data sample generated following a Gaussian distribution. These results demonstrate that our MeCo can achieve the highest correlation even using random data, which is fully data and label-independent.
>
> **Q6 (Other concerns in the *Weakness* part): The performance of other newer zero-cost proxies such as NASWOT, KNAS, NASI, and GradSign:**
>
> Answer: We would like to add the performance of these mentioned proxies as follows:
>
> NATSBench-TSS:
>
> || CIFAR-10 | CIFAR-100 | ImageNet16-120 |
> |-|-|-|-|
> | NASWOT| 0.77| 0.80| 0.77|
> | KNAS| 0.20| 0.35| 0.42|
> | NASI| 0.44| 0.43| 0.63|
> | GradSign| 0.77| 0.79| 0.78|
> | **MeCo (Ours)** | **0.89**| **0.88**| **0.85**|
>
> NATSBench-SSS:
>
> || CIFAR-10 | CIFAR-100 | ImageNet16-120 |
> |-|-|-|-|
> | NASWOT| 0.45| 0.43| 0.42|
> | KNAS| 0.25| 0.12| 0.32|
> | NASI| 0.17| 0.04| 0.20|
> | GradSign| 0.21| 0.16| 0.04|
> | **MeCo (Ours)** | **-0.80**| **-0.89**| **-0.88**|

---

### Official Review · Reviewer_gqC1 · 2023-07-07

**Soundness:** 3 good
**Presentation:** 2 fair
**Contribution:** 3 good
**Rating:** 6
**Confidence:** 5

**Summary:**

The paper proposes a new zero-cost proxy (called MeCo) for neural architecture search motivated by theoretical insights about the relationship of the statistics of Pearson Matrix and a network's convergence rate. The proposed proxy is evaluated by investigating ranking correlation on NATSBench-TSS, NATSBench-SSS, and NASBench101, as well as CIFAR1 dataset.

**Strengths:**

- A novel zero-cost proxy for NAS grounded in theory is proposed
- Good performance of the proposed method (although see below)

**Weaknesses:**

My concern primarily comes from the evaluation sections.
- The author didn't show the experiments on large-scale datasets, such as ImageNet
- The author didn't show the results on MobileNet-v2 like search space.
- The author didn't show the performance on other types of NAS benchmarks that are beyond the image classification, e.g., TransNASBench101.

I am also concerned about the theory part.
- The author only discusses the convergence speed of the network. The generalization capacity of the network is not involved in the discussion.
- The author shows contrasting results on TSS and SSS.





Minor:
- Typos: In line 220, the author claims that MeCo is 0.6-0.9 higher than ZiCo.

**Questions:**

My questions are aligned with the weakness sections
- How could Meco be connected with the generalization capacity of the network?
- Can the author show more results on diverse tasks beyond the image classification?
- How does the number of input samples impact the performance of the MeCo?
- Does MeCo generalize to other search spaces and large-scale datasets?
- Can the author show the performance of the searched optimal network via MeCo on these benchmarks?
- Why on SSS the correlation is negative while on TSS the correlation is positive? Based on the theoretical analysis (line 183, 184), it should be always positive.

**Limitations:**

Currently, the limitations are primarily the insufficient evaluation of different tasks and different search spaces. Also, some results are not clear in this paper.

---

> ### Author Rebuttal · Authors · 2023-08-08
>
> Thanks a lot for your time and valuable comments and suggestions! We would like to respond to your reviews point by point.
>
> **Q1: MeCo and generalization capacity:**
>
> Answer: MeCo can also reflect the network generalization capacity. Specifically, we introduce the following theorem:
>
> **Theorem 3.** For an over-parameterized neural network, we denote the loss as $L(W)$. Let $y=(y_1, ..., y_N)^T$ ,  $\lambda_0=\lambda_{min}(H^{\infty})$. Let $\gamma$ be the step of SGD, $\gamma = \kappa C_1\sqrt{y^T(H^\infty)^{-1}y}/(m\sqrt N)$ for some small enough absolute constant $\kappa$. Under the assumption of Theorem 1,  for any $\delta\in(0,e^{-1}])$, there exists $m^\ast(\delta,N,\lambda_0)$, such that if $m\geq m^*$, then with probability at least $1-\delta$, we have
> $$
> E[L(W)] \leq O(C'\sqrt{\frac{y^T y}{p_0 N}})+ O(\sqrt{\frac{\log(1/\delta)}{N}})
> $$
> where $C, C', \delta$ are constants.
> The proof can be obtained from Corollary 3.10 of [1] and Section D.2 of [2]. Theorem 3 indicates that the generalization capacity is bounded by $p_0$. The expectation of the loss decreases when $p_0$ gets larger. This relationship is consistent with Theorem 2.
>
> **Q2: More results on diverse tasks:**
>
> Answer: We conduct experiments on 10 tasks on TransBench-101-Micro and 7 tasks on Transbench-101-Macro to show the performance of MeCo. The results are shown in Tab 1 and 2 in the attached PDF.
>
> MeCo achieves the highest correlation on Surface Normal+Macro, Class Objection+Micro, SCifar-100+Micro, and NinaPro+Micro. MeCo also achieves competitive results on the majority of the remaining tasks. On some of the tasks, e.g., Autoencoding+Micro, MeCo only achieves 0.03 correlation. This might because the characteristics of the task per se, and the channel-sensitive trait of MeCo. Designing proxies that can be further generalized to more tasks is a promising direction. We would like to note that the existing proxies do not achieve a high correlation on all task consistently.
>
> **Q3: How does the #Samples impact the performance of MeCo?**
>
> Answer: We add ablation experiments about #Samples v.s. MeCo performance. We randomly chose different #Samples to evaluate the architectures in the entire TSS and SSS, respectively. The results are shown in Fig 1 in the attached PDF.
>
> It can be shown that MeCo is relatively stable when varying the #Samples. The correlation fluctuates around 0.01 throughout the experiments. These results support that MeCo only requires one data sample.
>
> **Q4: Does MeCo generalize to other search spaces and large-scale datasets?**
>
> Answer: 1) We add experiments on NASBench-301, NASBench-201 (with extra three datasets), transformer benchmark AutoFormer, and MobileNet-like search space (OFA). The results are shown in Tab 3 in the attached PDF.
>
> On the standard NASBench-301+CIFAR10 task, MeCo achieves the highest correlation with the model performance. On the three extra tasks, MeCo also obtains competitive results, e.g., 0.68 for SVHN, which is only 0.02 lower than SOTA baseline.
>
> We follow the settings in NASBench-Zero-Suite and adopt MeCo on three extra datasets on NASBench-201. The results show on SVHN, MeCo is 0.11 higher than the best baseline.
>
> * MobileNet-like search space (OFA)
>
> We follow the settings in OFA and utilize MeCo to evaluate the architectures. The correlation is **0.86**, which is relatively high.
>
> * Transformer search space (Autoformer)
>
> For AutoFormer, we load the trained supernets and regenerate the candidate subnets. We then re-evaluate the subnets to obtain the accuracy and compute MeCo. The correlation of MeCo and model accuracy on AutoFormer is **0.45**.
>
> 2）Large-scale datasets: We adopt MeCo-based NAS to ImageNet-1k on MobileNet-v2 search space. The results and comparisons are shown in Tab 4 in the attached PDF.
> We trained the searched optimal network with six RTX 3090 for about 130 hours. The Top-1 accuracy is **77.3\%**, which is higher than all the baseline approaches.
>
> **Q5: The searched optimal network:**
>
> Answer: We provide the accuracy of the searched optimal networks on four benchmarks, i.e., TSS/SSS with CIFAR10/CIFAR100/ImageNet, Darts with CIFAR10/CIFAR100, and MobileNetv2 with ImageNet-1k.
> The results of TSS/SSS/DARTS are shown in Tab 4, 5 in our paper, and Tab 4, 5, 6 in the appendix, with the visualized architecture structures in Fig 2.
> We add the experiment of Mobilenet-v2 with ImageNet-1k, the results are presented in the answer of Q4.
>
> **Q6: Negative correlation on SSS:**
>
> Answer: We have discussed this phenomenon in Section 4.2, p.7, ln. 235-246. More concretely, in our paper 3.2.3, we summarize that a multi-channel convolutional layer can be viewed as a multi-sample over-parameterized NN layer. This leads to the fact that MeCo is sensitive to #channels. The more channels contained, the larger the Pearson matrix, which will lead to SMALLER $p_0$. The simulated results in Fig 2 also validate this issue. On the other hand, larger #channels will normally lead to higher accuracy of the architectures (yet lower convergence rate). Thus, the variation of #channels in SSS  leads to the negative correlation on SSS. In TSS, however, all the networks share the same #channels at each stage, thus MeCo does not show negative correlation.
>
> We would like to further point out that this phenomenon does not contradict our theoretical basis. In practice, various factors (e.g., #channels, #samples) may affect the network accuracy. Theorem 2 and 3 show the relationship between $p_0$ and the convergence rate/generalization capacity of the network, which has been proven to be effective in multiple benchmarks.
>
> [1] Cao Y, Gu Q. Generalization bounds of stochastic gradient descent for wide and deep neural networks[J]. Advances in neural information processing systems, 2019, 32.
>
> [2] Zhu Z, Liu F, Chrysos G, et al. Generalization properties of NAS under activation and skip connection search[J]. Advances in Neural Information Processing Systems, 2022, 35: 23551-23565.

---

> > ### Comment · Reviewer_gqC1 · 2023-08-17
> >
> > I have some follow-up questions.
> >
> > 1. What's the model size of the obtained network on ImageNet? Could the author give more details such as FLOPs/MACs?
> > 2. I am not convinced by the explanation of the negative correlation on SSS. The problem is the author already prove that lower MeCo indicates a better network, which indicates a positive correlation.

---

> > > ### Author Response · Authors · 2023-08-18
> > > **Response to the Follow-up Questions**
> > >
> > > Dear Reviewer gqC1,
> > >
> > > Thanks for your valuable questions! We would like to answer as follows.
> > >
> > > Q1: We use Zero-Cost-PT+MeCo to search the networks on the MobileNet-V2 search space. The FLOPs of the obtained network is 884M. We did not limit the upper bound of FLOPs because Zero-Cost-PT is designed to be unconstrained NAS, which has no limits on FLOPs.
> > >
> > > Q2: Theorem 2 clarifies that a larger MeCo leads to a larger **convergence rate**, not a higher accuracy. Similarly, Theorem 3 says that **when the number of channels is fixed (i.e., $N$)**, the larger the Meco, the better the **generalization capacity**. In fact, it is not easy to directly analyze the network accuracy. Thus instead, works in this region utilize the convergence rate and generalization capacity to evaluate the network performance. In this paper, we follow the settings in [1] and fix $N$ to see the impact of $p_0$ on the network performance.
> > >
> > > However, on SSS, $N$ is not a fixed value. Specifically, the accuracy of the networks has a strong positive correlation with the number of channels (equivalent to $N$ in our case, Section 3.2.3), which is determined by the benchmark per se. Simultaneously, the increase in the number of channels leads to a **slower convergence rate**, thus leading to smaller MeCo. In fact, $N$ has an opposite influence on convergence rate and generalization capacity. Such a negative correlation is consistent with theoretical and empirical analysis.
> > >
> > > Finally, the strong negative correlation derives from MeCo viewing a multi-channel CNN layer as a multi-sample NN layer. **Benefiting from this claim, MeCo only requires one data sample, yet leading to the channel-sensitivity.** There are also previous works that show opposite results across different benchmarks, e.g., grasp on TSS and SSS, gradnorm on TSS and TransBench101. This is determined by the construction of the proxies and the benchmark characteristics.
> > >
> > > [1] Cao Y, Gu Q. Generalization bounds of stochastic gradient descent for wide and deep neural networks[J]. Advances in neural information processing systems, 2019, 32.

---

> > > ### Author Response · Authors · 2023-08-21
> > > **Sincerely expecting further discussions with reviewer gqC1**
> > >
> > > Dear Reviewer gqC1,
> > >
> > > Thanks for your constructive suggestions in your review. Since the discussion period is ending soon, have our previous responses addressed your concerns?
> > >
> > > We would be more than happy to further discuss with you and provide more information.
> > >
> > > Best,
> > > Authors, Paper 7200

---

> > > > ### Comment · Reviewer_gqC1 · 2023-08-21
> > > >
> > > > Dear authors,
> > > >
> > > > Thanks for your response. I also read your response to other reviewers. My concern has been solved. I increase my rating from 4 to 6.

---

> > > > > ### Author Response · Authors · 2023-08-21
> > > > > **Thanks for increasing the score!**
> > > > >
> > > > > Dear Reviewer gqC1,
> > > > >
> > > > > We truly appreciate your valuable suggestions! Thanks for increasing the rating!
> > > > >
> > > > > Best,
> > > > >
> > > > > Authors 7200

---

### Author Rebuttal · Authors · 2023-08-09

General Response to the Reviewers.

We thank all the reviewers for their time and valuable comments and suggestions, which are indeed promising and can improve the quality of our paper! In this rebuttal period, we made several improvements and additions to the current version in terms of both theoretical and experimental aspects.

We summarize the main improvements here and reply to each of the reviewers respectively.

* Experimental Aspects

    - Performance of MeCo on more benchmarks (Rev1, Rev3, Rev4, Rev5): We add experiments on NASBench-301, NASBench-201 (with extra three datasets), and MobileNet-like benchmark (OFA) to show the effectiveness of MeCo.
    - Performance of MeCo on other tasks and regions (Rev1, Rev3, Rev4): We add experiments on Transbench-101-Macro, Transbench-101-Micro (including extra three datasets), and a transformer benchmark (AutoFormer) to show the performance of MeCo for various tasks.
    - Performance of MeCo on large-scale dataset ImageNet-1k (Rev1, Rev4, Rev5): Other than the OFA benchmark (which is trained on ImageNet), we also add experiments on ImageNet-1k with MobileNet v2 and provide the Top-1 accuracy of the selected architecture.
    - Impact of #Samples on MeCo (Rev1, Rev4): We conduct a new ablation study in terms of the impact of #samples on MeCo.
    - Variation and robustness of MeCo (Rev2, Rev4): We provide the variation of MeCo for different #Samples from the same or different classes.
    - Missing baselines and evaluation metrics (Rev2, Rev3): We add the performance of the baselines and provide the time cost comparison.

* Theoretical Aspects

    - Generalization capacity of MeCo (Rev1, Rev5): We provide a new theorem to show the connection between MeCo and generalization capacity.
    - Negative correlation on SSS (Rev1, Rev3, Rev4): We elaborate more on this issue respectively corresponding to each reviewer's question.

* Clarity

  We clarify the following issues regarding the reviewer's confusion:

    - Name of the benchmarks (Rev2).
    - Goal of the ablation study (Rev4, Rev5).
    - Inconsistent results of NTK (Rev2).
    - Time cost in Table 5 (Rev5).
    - Accuracy of the selected optimal architecture (Rev1).

* Other Improvements

  - ``Limitations`` section (Rev3): We thank the reviewer for the promising insight. We elaborate more on these issues and will add them to the paper.
  - ``Related Work`` section (Rev4): We will add the papers as references and elaborate more.
  - Typo in ln.220 (Rev1, Rev5): We will fix the typo.

We present the experimental results on Transbench-101-Micro, Transbench-101-Macro, NASBench301, NASBench201 (extra tasks), and model accuracy on ImageNet-1k in the attached PDF due to the character limits.

---

### Author Response · Authors · 2023-08-21
**General response regarding the channel-sensitive issue of MeCo**

We appreciate all the valuable comments from the reviewers. During the rebutal and discussion phases, we have proposed an optimization method (i.e., MeCo-opt) to alleviate the negative correlation issue. We would like to present the detailed method and experimental results here as a general response.

1.  MeCo-opt

For the feature map of layer $l$, the original MeCo flattens each channel of the feature map and computes $p_0$ as the evaluation result. This operation makes MeCo be sensitive to \#channels, e.g., the larger \#channels, the smaller $p_0$, even $p_0=0$. To address this issue, alternatively, we randomly **sample a fixed number of channels** and flatten them to compute $p_0'$. We then compute the final result by **multiplying $p_0'$ with a channel weight**, such that for layer $l$,

$$
\mathsf{MeCo}_{\mathsf{opt}}^{l} := \frac{c}{n} \cdot p_0'
$$

where $c$ is #channels, $n$ is the fixed sampling numbers.

The high-level idea of this optimization is to limit the dimension of the Pearson correlation matrix by constraining #channels. Instead of computing upon a large matrix, we calculate the minimum eigenvalue upon a fixed-sized matrix and then enlarge it with corresponding constants. Note that the bonus of this optimization is that the time-cost can be controlled because the matrix dimension is significantly reduced compared to the original one.

2. Performance of MeCo-opt

In the following experiments, we set $n=8$.

* Correlation on TSS and SSS

|| Cifar-10| Cifar-100| ImageNet16-120|
|-|-|-|-|
| TSS | 0.90 $\pm$ 0.002 | 0.89 $\pm$ 0.003 | 0.85 $\pm$ 0.003 |
| SSS | 0.89 $\pm$ 0.002 | 0.83 $\pm$ 0.004 | 0.89 $\pm$0.003|

* Correlation on NASBench-301

|Method|Cifar-10|Spherical-Cifar-100|NinaPro|SVHN|
|-|-|-|-|-|
|grasp|0.34|**0.13**|0.04|0.18|
|fisher|-0.28|0.00|-0.11|0.05|
|grad_norm|-0.04|-0.00|**-0.20**|0.42|
|snip|-0.05|-0.01|-0.10|0.38|
|synflow|0.18|0.05|-0.07|0.50|
|l2_norm|0.45|0.12|-0.07|**0.70**|
|#params|0.46|0.07|-0.07|**0.70**|
|zen|0.43|0.07|-0.09|0.68|
|jacov|-0.04|0.08|0.13|-0.36|
|nwot|0.47|0.05|0.02|0.64|
|zico|0.66|-|-|-|
|MeCo (Ours)|0.70|-0.05|-0.11|0.68|
|**MeCo-opt**|**0.71**|**0.03**|**0.12**|**0.68**|

* * Transbench101-Macro

|Method|Autoencoder|Class Objection|Scene Classification|Jigsaw|Surface Normal|Segmantation|Room Layout|
|-|-|-|-|-|-|-|-|
|grasp|-0.02|-0.64|-0.43|-0.26|-0.05|-0.02|-0.26|
|fisher|-0.19|-0.30|-0.13|-0.26|0.15|0.03|-0.26|
|grad_norm|0.31|-0.56|-0.33|-0.27|0.35|0.21|-0.27|
|snip|0.20|-0.38|-0.14|-0.19|0.45|0.27|-0.19|
|synflow|0.00|0.12|0.27|0.34|0.00|0.00|0.34|
|l2_norm|-0.20|0.08|0.28|0.15|0.30|0.18|0.15|
|#params|-0.18|0.16|0.32|0.15|0.30|0.06|0.15|
|zen|-0.01|0.10|0.27|0.24|0.38|0.27|0.24|
|jacov|0.45|0.07|0.19|0.19|0.50|0.57|0.19|
|nwot|0.67|**0.83**|**0.89**|0.48|0.78|**0.80**|**0.76**|
|MeCo (Ours)|0.51|0.59|0.81|0.17|0.80|0.62|0.23|
|**MeCo-opt**|**0.74**|**0.73**|**0.76**|**0.48**|**0.76**|**0.63**|**0.33**|

* Transbench101-Micro

|Method|Autoencoding|Class Objection|Scene Classification|Jigsaw|Surface Normal|Segman- tation|Room Layout|Spherical-Cifar-100|NinaPro|SVHN|
|-|-|-|-|-|-|-|-|-|-|-|
|grasp|-0.12|-0.22|-0.43|-0.12|0.01|0.00|-0.29|-0.03|-0.20|-0.24|
|fisher|**-0.58**|0.44|-0.13|0.30|0.16|0.12|0.30|0.72|0.42|0.81|
|grad_norm|-0.32|0.39|-0.33|0.36|0.36|0.60|0.25|0.72|0.40|0.78|
|snip|-0.27|0.45|-0.14|0.41|0.49|0.68|0.32|0.76|0.42|0.83|
|synflow|0.00|0.48|0.27|0.47|0.00|0.00|0.30|0.79|0.45|**0.92**|
|l2_norm|0.04|0.32|0.28|0.35|0.50|0.48|0.18|0.53|0.36|0.52|
|#params|-0.01|0.45|0.32|0.44|0.62|0.68|0.30|0.79|0.36|0.76|
|zen|0.14|0.54|0.27|0.51|0.71|0.67|0.38|0.67|0.42|0.74|
|jacov|0.18|0.51|0.19|**0.56**|**0.75**|**0.80**|**0.40**|0.71|0.40|0.77|
|nwot|0.03|0.39|**0.89**|0.42|0.57|0.53|0.25|0.64|0.38|0.63|
|zico|0.35|-|0.71|0.52|0.68|-|-|-|-|-|
|MeCo (Ours)|0.03|0.58|0.62|0.45|0.65|0.62|-0.25|0.85|0.47|0.88|
|**MeCo-opt**|**0.03**|**0.59**|**0.64**|**0.47**|**0.67**|**0.77**|**0.26**|**0.85**|**0.47**|**0.88**|

* Accuracy on TSS

|| Cifar-10| Cifar-100| ImageNet16-120|
|-|-|-|-|
|ZiCo| 93.56 $\pm$ 0.23 |70.64 $\pm$ 0.28 | 42.74 $\pm$ 1.78|
|synflow| 93.27 $\pm$ 0.74 |**71.12 $\pm$ 1.59** | 42.65 $\pm$ 3.59|
|MeCo|**93.64 $\pm$ 0.31**| 70.86 $\pm$ 0.96| 42.59 $\pm$ 1.77|
|MeCo-opt| 93.63 $\pm$ 0.26 | 70.79 $\pm$ 0.35 | **42.84 $\pm$ 1.60**|

* Accuracy on SSS

|| Cifar-10| Cifar-100| ImageNet16-120|
|-|-|-|-|
|ZiCo| 93.06 $\pm$ 0.23 | 69.65 $\pm$ 0.54 | 45.40 $\pm$ 0.72|
|MeCo| 93.12 $\pm$ 0.24 | **69.89 $\pm$ 0.42** | 44.34 $\pm$ 1.78|
|MeCo-opt| **93.37 $\pm$ 0.19** | 69.75 $\pm$ 0.35 | **46.62 $\pm$ 0.70**|

In all, MeCo-opt effectively alleviates the channel-sensitive issue of MeCo. We will add a ``Discussion and Optimization`` section and elaborate these results in detail.

---

### Decision · Program_Chairs · 2023-09-21

**Decision:**

Accept (spotlight)

**Comment:**

Reviewers were enthusiastic about the paper and recognized a technically sound paper with good results. Reviewers raise some minor point in their review which have been addressed in the rebuttal. We invite the authors to take these changes into account in the camera-ready version of the paper.